# Open Problems in Technical AI Governance

**Anka Reuel**[*1]                                          *anka.reuel@stanford.edu*
**Ben Bucknall**[*2,3]                                       *bucknall@robots.ox.ac.uk*

**Stephen Casper**[4], **Tim Fist**[5,6], **Lisa Soder**[7], **Onni Aarne**[8], **Lewis Hammond**[2,9], **Lujain Ibrahim**[2],
**Alan Chan**[10,11], **Peter Wills**[2,10], **Markus Anderljung**[10], **Ben Garfinkel**[10], **Lennart Heim**[10],
**Andrew Trask**[2,12], **Gabriel Mukobi**[1], **Rylan Schaeffer**[1], **Mauricio Baker**[13], **Sara Hooker**[14],
**Irene Solaiman**[15], **Alexandra Sasha Luccioni**[15], **Nitarshan Rajkumar**[16], **Nicolas Moës**[17],
**Jeffrey Ladish**[18], **David Bau**[19], **Paul-Andrei Bricman**[20], **Neel Guha**[1], **Jessica Newman**[21],
**Yoshua Bengio**[11,22], **Tobin South**[23], **Alex Pentland**[24], **Sanmi Koyejo**[1,25], **Mykel J. Kochenderfer**[1],
**Robert Trager**[2,3]

[1]*Stanford University* [2]*University of Oxford* [3]*Oxford Martin AI Governance Initiative* [4]*MIT CSAIL*
[5]*Institute for Progress* [6]*Center for a New American Security*
[7]*interface - Tech Analysis and Policy Ideas for Europe e.V.* [8]*Institute for AI Policy and Strategy*
[9]*Cooperative AI Foundation* [10]*Centre for the Governance of AI* [11]*Mila* [12]*OpenMined* [13]*Independent Researcher*
[14]*Cohere for AI* [15]*Hugging Face* [16]*University of Cambridge* [17]*The Future Society* [18]*Palisade Research*
[19]*Northeastern University* [20]*Noema Research* [21]*University of California, Berkeley* [22]*University of Montreal* [23]*MIT*
[24]*Stanford HAI* [25]*Virtue AI*

*Reviewed on OpenReview:* *https://openreview.net/forum?id=1nO4qFMiS0*

## Abstract

AI progress is creating a growing range of risks and opportunities, but it is often unclear how they should be navigated. In many cases, the barriers and uncertainties faced are at least partly technical. Technical AI governance, referring to technical analysis and tools for supporting the effective governance of AI, seeks to address such challenges. It can help to (a) identify areas where intervention is needed, (b) assess the efficacy of potential governance actions, and (c) enhance governance options by designing mechanisms for enforcement, incentivization, or compliance. In this paper, we explain what technical AI governance is, outline why it is important, and present a taxonomy and incomplete catalog of its open problems. This paper is intended as a resource for technical researchers or research funders looking to contribute to AI governance.

---

*Equal contribution; corresponding authors; order randomized. Work completed while BB was at Centre for the Governance of AI.

Given its scope, inclusion as an author does not entail endorsement of all aspects of the paper, with the exception of AR and BB.

Cite as Reuel, Bucknall, et al. (2025) "Open Problems in Technical AI Governance."

# 1   Introduction

The rapid development and adoption of artificial intelligence (AI) systems[1] has prompted a great deal of governance action from the public sector,[2] academia and civil society (Anderljung et al., 2023a; Moës & Ryan, 2023; Barrett et al., 2023), and industry (Anthropic, 2023a; Microsoft, 2023; Dragan et al., 2024; OpenAI, 2024a), with the aim of addressing potential risks while capitalizing on benefits.

However, key decision-makers seeking to govern AI often have insufficient information for identifying the need for intervention and assessing the efficacy of different governance options. Furthermore, the technical tools necessary for successfully implementing governance proposals are often lacking (Reuel et al., 2024a), leaving uncertainty regarding how policies are to be implemented. For example, while the concept of watermarking[3] AI-generated content has gained traction among policymakers (see for example Council of the European Union, 2024; The White House, 2023b; G7 leaders, 2023; Department for Science, Innovation & Technology, 2023), it is unclear whether current methods are sufficient for achieving policymakers' desired outcomes, nor how future-proof such methods will be to improvements in AI capabilities (Zhang et al., 2023; Ghosal et al., 2023). Addressing these and similar issues will require further targeted technical advances.

In this paper, we aim to catalyse work aimed at addressing such challenges and uncertainties through motivating and introducing the field of **technical AI governance (TAIG)**, providing a taxonomy of subareas, and outlining open problems and research questions. Throughout the paper we use the term technical AI governance to refer to *technical analysis and tools for supporting the effective governance of AI*, where 'technical' pertains to the physical sciences, mathematics, engineering, and related fields; and 'AI governance' refers to the processes and structures through which decisions related to AI are made, implemented and enforced.[4] AI governance includes research that is both non-technical (e.g. in politics, economics, law, philosophy, etc.) and technical in nature (e.g. in computer science, engineering, and mathematics). Through our use of this definition we simply refer to this latter category of work within the umbrella category of 'AI governance.'

By the above definition, TAIG can contribute to AI governance in a number of ways, such as by identifying opportunities for governance intervention, informing key decisions, and enhancing options for implementation. For example, deployment evaluations that assess the downstream impacts of a system (see Section 3.4) could help identify a need for policy interventions to address these impacts. Alternatively, being able to design models that are robust to malicious modifications (see Section 6.4) could add to the menu of governance options available to prevent downstream misuse.

In particular, we make the following contributions:

- We introduce the field of technical AI governance and motivate the need for work in this area.

- We present a taxonomy of TAIG arranged along two dimensions: *capacities*, which refer to actions such as access and verification that are useful for governance, and *targets*, which refer to key elements in the AI value chain, such as data and models, to which capacities can be applied.

- Finally, we outline open problems within each category of our taxonomy, along with concrete example questions for future research.

---

[1]Our understanding of AI systems follows that of (Basdevant et al., 2024), encompassing infrastructure such as compilers, model components such as datasets, code, and weights, as well as UX considerations.

[2]See, for example, (The White House, 2023a; The White House Office of Science and Technology Policy, 2023; Presidency of the Council of the European Union, 2024; Department for Science, Innovation and Technology et al., 2023a; Department for Science, Innovation and Technology & Office for Artificial Intelligence, 2023; Advisory Body on Artificial Intelligence, 2023; European Commission, 2023; 国家互联网信息办公室国家发展和改革委员会教育部等, 2023)

[3]Watermarks are signals placed in output content that are imperceptible to humans, but easily detectable through application of a specific *detector* algorithm.

[4]To the extent that this definition of TAIG includes measures for directly increasing the performance, safety, or robustness of AI systems, we only consider such measures for cases in which they support the governance of AI.

Figure 1 provides an overview of the open problem areas, organized according to the taxonomy. We hope that this paper serves as a resource and inspiration for technical researchers aiming to direct their expertise towards policy-relevant topics.

## 1.1 Relation to AI Governance

As noted above, we define AI governance as the processes and structures through which decisions related to AI are made, implemented, and enforced. It encompasses the rules, norms, and institutions that shape the behavior of actors in the AI ecosystem, as well as the means by which they are held accountable for their actions.[5] As per our definition above, TAIG consists of *technical analysis and tools for supporting the effective governance of AI.* Here we outline three ways in which TAIG can contribute to AI governance, which we refer to as *identifying*, *informing*, and *enhancing*.[6]

Firstly, **TAIG can identify areas where governance intervention is needed**, through mapping technical aspects of AI systems to social and political concepts, typically conceived of as being addressed through governance. For example, tracking and considering technical advances in AI video generation could allow for more accurate predictions of the risk of video deepfakes, and thus motivate the need for a governance response.

Secondly, **TAIG can inform governance decisions** by providing decision-makers with more accurate information, allowing them to better compare the effectiveness of different governance options. For example, policymakers can choose between different regulatory instruments (for example, registration or disclosure), as well as how they enforce compliance (for example, *ex ante* rules or *post hoc* adjudication), with the efficacy of these options potentially depending on technical details. Information could stem from implemented TAIG methods, such as the outcome of assessments (see Section 3), or TAIG research that maps or monitors the AI ecosystem (see Section 8). For example, more developed risk models for assessing potential harms of AI could inform targeted policies for their mitigation.

Finally, **TAIG can enhance governance options** by providing or enabling mechanisms for enforcing, incentivizing, or complying with mandated requirements. For instance, developing methods for the robust evaluation of models with black-box access could facilitate more comprehensive third-party auditing, thereby enhancing enforcement of safety requirements.

Previous overviews of AI governance, such as Dafoe (2018)'s AI governance research agenda, focus primarily on socio-political governance challenges, including labor displacement, inequality, shifts in national power, and AI race dynamics, or taxonomizes themes, gaps, and issues in AI governance more broadly (Birkstedt et al., 2023). This literature sets the stage for why technical solutions are needed (e.g., an "AI race that sacrifices safety" (Dafoe, 2018) could imply a need for governance-oriented technical security mechanisms). More recent policy documents (see e.g., OECD, 2019; G7, 2023) enumerate high-level issues like privacy concerns, bias, a lack of accountability, and security in AI, while including technical, socio-political, and institutional considerations. Our work contributes by focusing more narrowly on the technical dimension of those issues. For instance, where such reports highlight AI accountability as a governance challenge, our taxonomy offers concrete technical problem areas (like verifiable auditing and data provenance tracking) that would help achieve accountability in practice.

The AI risk literature categorizes both realized harms and anticipated risks of AI (see e.g., Lobel, 2024; Shelby et al., 2023; Raji et al., 2022a; Weidinger et al., 2022; Critch & Russell, 2023). For example, Shelby et al. (2023) presented a human-centered taxonomy of sociotechnical harms from algorithmic systems, identifying five harm categories (representational, allocative, quality-of-service, interpersonal, and societal harms) based on a review of 172 research papers. Similarly, Uuk et al. (2024) conducted a systematic literature review to derive 13 categories of systemic AI risks, including environmental harms, structural discrimination, and loss of control. Similar to the literature on challenges in AI governance, this prior work is relevant to our efforts in that it motivates work on AI governance – indeed, many of the open problems we list (such as developing robust evaluation methods) are motivated by those very risks. By situating our work among these prior

---

[5]For other proposed definitions of AI governance, see (Bullock et al., 2022; Daly et al., 2021; Dafoe, 2018).

[6]A useful parallel to TAIG may be the concepts of regulatory technology (RegTech) and supervisory technology (SupTech) in the financial sector (Bank for International Settlements, 2021), which aim to support financial regulation and oversight.

## Taxonomy of Open Problems Areas in Technical AI Governance

| | Data | Compute | Models and Algorithms | Deployment |
|---|---|---|---|---|
| **Assessment** | • Identification of Problematic Data
• Infrastructure and Metadata to Analyze Large Datasets
• Attribution of Model Behaviour to Data | • Definition of Chip and Cluster Specifications for Model Training
• Classification of Workloads | • Reliable Evaluations
• Efficient Evaluations
• (Multi-)Agent Evaluations | • Downstream Impact Evaluations |
| **Access** | • Privacy-Preserving Third-Party Access to Datasets
• Preservation of Evaluation Data Integrity | • Provision of Compute Resources | • Facilitation of Third-Party Access to Models | • Access to Downstream User Logs and Data |
| **Verification** | • Verification of Training Data | • Verification of Chip Location
• Verification of Compute Workloads | • Verification of Model Properties
• Verification of Dynamic Systems
• Proof of Learning | • Verifiable Audits
• Verification of AI-generated Content |
| **Security** | • Detection and Prevention of Training Data Extraction | • Use of Hardware Mechanisms for AI Security
• Anti-Tamper Hardware
• Enforcement of Compute Usage Restrictions | • Prevention of Model Theft
• Shared Model Governance
• Model Disgorgement and Machine Unlearning | • Detection of Adversarial Attacks
• Modification-Resistant Models
• Detection and Authorization of Dual-Use Capability at Inference Time |
| **Operationalization** | • Translation of Governance Goals into Policies and Regulatory Requirements
• Deployment Corrections | | | |
| **Ecosystem Monitoring** | • Clarification of Associated Risks
• Prediction of Future Developments and Impacts
• Assessment of Environmental Impacts
• Supply Chain Mapping | | | |

Figure 1: An overview of the open problem areas covered in this report, organized according to our taxonomy.

catalogs, we underscore that our contribution is a focused extension: we zoom in on the technical layer of AI governance to provide a structured overview of open technical problems that underlie or intersect with identified AI governance challenges.

## 1.2 Scope and Limitations

This paper aims to give a broad overview of open technical problems for AI governance, identifying gaps in existing or suggested governance proposals, while avoiding taking a normative position on their desirability or efficacy. Indeed, the governance aims motivating some of the open questions outlined below may be in tension with each other, and we do not expect their solutions all to be used within the same governance framework. For example, broad access to some AI systems may be in conflict with ensuring their security.

At the same time, we are conscious of the potential pitfalls of techno-solutionism – that is, relying solely on proposed technical fixes to complex and often normative social problems – including a lack of democratic oversight and introducing further problems to be fixed (Michael et al., 2020; Lindgren & Dignum, 2023; Angel & Boyd, 2024; Allen, 2024). Many of the TAIG tools presented below are hypothetical and speculative, and we make no claims about the feasibility of developing solutions. Furthermore, some of the TAIG measures highlighted are dual-use. For example, while hardware-enabled mechanisms for monitoring advanced compute hardware could provide increased visibility into the private development of the largest models, they could also potentially be applied to unreasonably surveil individuals using such hardware for legitimate purposes.

Thus, having solutions to all open problems outlined in this paper will not have *solved* AI governance. On the contrary, careful management will be necessary to determine a balance between capacities that are in tension with each other, and to ensure that dual-use capacities are not misused. Furthermore, many AI governance problems may rely predominantly on non-technical solutions, such as ensuring the appropriate inclusion of countries impacted by AI in international AI governance decision-making (Trager et al., 2023). However, we argue that making progress on the technical problems outlined below can help to ensure more robust AI governance on net.

We view TAIG as related to (and in some places overlapping with) existing areas of research. Namely, TAIG differs from topics in AI safety and alignment in that it is not aimed at directly improving the safety of AI systems. Furthermore, while sociotechnical approaches to AI safety and governance (see e.g. Lazar & Nelson, 2023; Bogen & Winecoff, 2024; Oduro & Kneese, 2024) view "society and technology together as one coherent system" (Chen & Metcalf, 2024), TAIG considers the instrumental value of technical work for enacting governance. Taken together with TAIG, these diverse approaches can serve as complementary methods for mitigating risks and promoting beneficial outcomes of AI (Narayanan et al., 2023). Finally, and as noted above, we view TAIG as a subset of AI governance more broadly, which also includes many topics in diverse disciplines including political science, economics, and philosophy.

We consider some notable fields, topics, and problems to be out of scope for this paper. In particular, technical work that directly improves the performance, safety, or robustness of AI systems, or addresses related ethical concerns – while highly relevant to AI governance – is considered out of scope. Topics regarding government or public-sector use of AI (Margetts & Dorobantu, 2019; Aitken et al., 2022; Margetts, 2022; Straub et al., 2023) or ways in which AI could itself be used to defend against or ameliorate downstream harms of AI (Bernardi et al., 2024) are also out of scope.

### 1.3 Reader Guide

Table 1: Relevant problem areas organized by reader background

| | | |
|---|---|---|
| **ML Theory** | Assessment | 3.1.2; 3.1.3; 3.2.1; 3.2.2; 3.3.1; 3.3.2; 3.4.1 |
| | Access | 4.1.1; 4.2.1; 4.3.1 |
| | Verification | 5.1.1; 5.2.2; 5.3.1; 5.3.3; 5.4.1; 5.4.2 |
| | Security | 6.1.1; 6.3.1; 6.3.2; 6.3.3; 6.4.1; 6.4.2; 6.4.3 |
| | Operationalization | 7.2 |
| **Applied ML** | Assessment | 3.1.2; 3.3.1; 3.3.2; 3.4.1 |
| | Access | 4.3.1; 4.4.1 |
| | Security | 6.4.3 |
| | Operationalization | 7.1; 7.2 |
| | Ecosystem Monitoring | 8.1; 8.2; 8.3 |
| **Cybersecurity** | Verification | 5.2.2 |
| | Security | 6.2.1; 6.2.3; 6.3.1; 6.4.3 |
| | Operationalization | 7.2 |
| **Cryptography** | Assessment | 3.1.1; 3.2.2 |
| | Access | 4.1.1; 4.1.2; 4.2.1; 4.3.1; 4.4.1 |
| | Verification | 5.1.1; 5.2.1; 5.2.2; 5.3.3; 5.4.1; 5.4.2 |
| | Security | 6.2.1; 6.2.3; 6.3.2; 6.4.3 |
| **Hardware Engineering** | Assessment | 3.1.2; 3.2.1; 3.2.2 |
| | Access | 4.2.1 |
| | Verification | 5.2.1; 5.2.2 |
| | Security | 6.2.1; 6.2.2; 6.2.3; 6.3.1; 6.3.2 |
| **Software Engineering** | Assessment | 3.1.1; 3.1.2; 3.3.2; 3.4.1 |
| | Access | 4.2.1 |
| | Verification | 5.2.2 |
| | Security | 6.2.1; 6.3.1 |
| **Mathematics and Statistics** | Assessment | 3.1.2; 3.4.1 |
| | Ecosystem Monitoring | 8.2; 8.3 |

This paper provides a broad overview of open problems across the taxonomy defined in Section 2. Given the extensive nature of the main content (Sections 3-8), we have structured it for selective reading:

- **Each section is self-contained**, allowing readers to focus on their area(s) of interest.

- Each section begins with a **summary table of problem areas**.

- **Specific open problems** within each area are highlighted in **bold**.

- **Example research questions** are provided in boxes at the start of each subsection.

- Table 1 offers suggested relevant problem areas based on **reader expertise**.

- We attach a two-page **policy brief** in appendix A.

This structure aims to facilitate quick identification of key issues and relevant problems for readers across various backgrounds and interests.

## 2 Taxonomy

The paper is organized according to a two-dimensional taxonomy, based around capacities and targets. *Capacities* encompass a comprehensive suite of abilities and mechanisms that enable stakeholders to understand and shape the development, deployment, and use of AI, such as by assessing or verifying system properties. These capacities are neither mutually exclusive nor collectively exhaustive, but they do capture what we believe are the most important clusters of technical AI governance. We list all considered capacities, along with descriptions, in Table 2. An overview of our methodology, and in particular how we derived this taxonomy, can be found in Appendix B.

Table 2: Overview of capacities and their importance for AI governance

| Capacity | Description | Why it matters for governance |
|---|---|---|
| Assessment | The ability to evaluate AI systems, involving both technical analyses and consideration of broader societal impacts. | Enables the identification and understanding of system capabilities and risks, allowing for more targeted governance intervention. |
| Access | The ability to interact with AI systems, including model internals, as well as obtain relevant data and information while avoiding unacceptable privacy costs. | Enables external research and assessment of AI systems, and aids in fairly distributing benefits of AI across society. |
| Verification | The ability of developers or third parties to verify claims made about AI systems' development, behaviors, capabilities, and safety. | Establishes trust in AI systems and confirms compliance with regulatory requirements. |
| Security | The development and implementation of measures to protect AI system components from unauthorized access, use, or tampering. | Ensures the integrity, confidentiality, and availability of AI systems and guards against misuse. |
| Operationalization | The translation of ethical principles, legal requirements, and governance objectives into concrete technical strategies, procedures, or standards. | Bridges the gap between abstract principles and practical implementation of regulatory requirements. |
| Ecosystem Monitoring | Understanding and studying the evolving landscape of AI development and application, and associated impacts. | Enables informed decision-making, anticipation of future challenges, and identification of key leverage points for effective governance interventions. |

The second axis of our taxonomy pertains to the *targets* that encapsulate the essential building blocks and operational elements of AI systems[7] that governance efforts may aim to influence or manage. They are adapted from categories introduced in (Bommasani et al., 2023b). Each capacity given above can be applied to each target. We structure our paper around the resulting pairs of capacities and targets, with the exception of *operationalization* and *ecosystem monitoring* which cut across all targets. The targets considered in this report are summarized in Table 3.

We recognize that organizational processes undertaken during the development and deployment of AI systems intersect with and shape these targets, and could be considered as regulatory targets in their own right.

---

[7](Repeat of footnote 1) Our understanding of AI systems follows that of (Basdevant et al., 2024), encompassing infrastructure such as compilers, model components such as datasets, code, and weights, as well as UX considerations.

Table 3: Overview of targets

| Target | Description |
| --- | --- |
| Data | The pretraining, fine-tuning, retrieval, and evaluation datasets on which AI systems are trained and benchmarked. |
| Compute | Computational and hardware resources required to develop and deploy AI systems. |
| Models and Algorithms | Core components of AI systems, consisting of software for training and inference, their theoretical underpinnings, model architectures, and learned parameters. |
| Deployment | The use of AI systems in real-world settings, including user interactions, and the resulting outputs, actions, and impacts. |

However, we have chosen not to include them as explicit targets in our taxonomy as processes mostly involve non-technical challenges that fall outside the scope of our paper. In cases where processes do face challenges, we address such issues within the context of the most relevant target. For example, compliance with content-creators' right to opt-out is dependent on identifying copyrighted samples in datasets (Section 3.1).

# 3  Assessment

Evaluations and assessments of the capabilities and risks of AI systems have been proposed as a key component in AI governance regimes. For example, *model evaluations and red-teaming*[8] comprised a key part of the voluntary commitments agreed between labs and the UK government at the Bletchley Summit (Department for Science, Innovation & Technology, 2023). Furthermore, the White House Executive Order on Artificial Intelligence requires developers of the most compute-intensive models to share the results of all red-team tests of their model with the federal government (The White House, 2023a).

The purpose of assessment is to detect problematic behavior or impacts of AI systems before resulting harms can materialize, as well as to ensure systems are safe, robust, and non-discriminatory. However, the assessment of some targets, especially in the context of foundation models, is currently more an art than a science, with a significant number of open challenges (Chang et al., 2024; Weidinger et al., 2023). These issues are exacerbated by the fact that evaluations are expensive to conduct at scale. While assessment and evaluation standards are emerging (National Institute of Standards and Technology (NIST), 2023; UK AI Safety Institute, 2024), there are still fundamental open technical problems that need to be addressed to ensure robust and informative assessments.

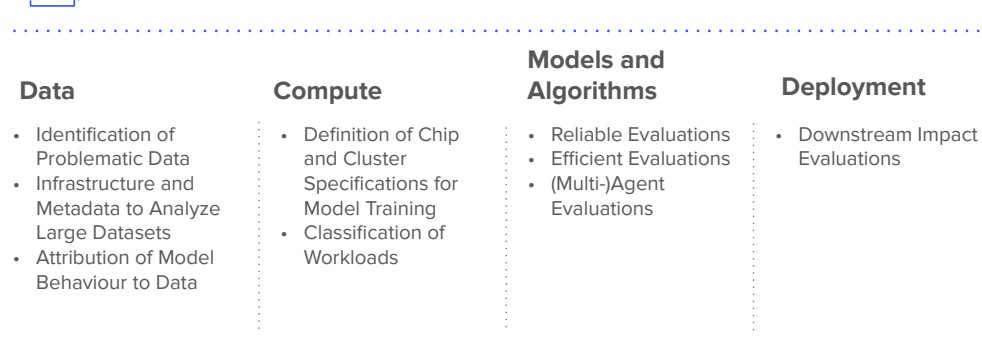

Figure 2: Open problem areas in the *Assessment* capacity, organized by target

## 3.1  Data

| Example Research Questions |
| --- |
| 1. How can methods for identifying problematic data be scaled to large (on the magnitude of trillions of tokens/samples) datasets? (3.1.1) |
| 2. How can license collection be automated to prevent training on unlicensed data? (3.1.1) |
| 3. How can the accuracy of licenses be ensured when aggregating datasets from multiple sources? (3.1.1) |
| 4. How can problematic data be identified without full/direct access to the dataset? (3.1.1) |
| 5. How can contamination of training data with problematic samples be reliably detected? (3.1.1) |
| 6. How can harmful data be removed from a dataset without facilitating its easy identification by malicious actors? (3.1.1) |

---

[8]*Red-teaming* refers to deliberately trying to find ways to make a system behave poorly, produce harmful outputs, or be misused, in order to identify potential risks and vulnerabilities to be addressed.

7. What license and meta-data reporting requirements could assist in responsible data practices? (3.1.2)

8. What infrastructure is needed to enable researchers to audit large datasets? (3.1.2)

9. How can macro-scale dataset properties, such as persistent bias, be identified and measured? (3.1.2)

10. What information about datasets is necessary for determining their suitability for training? (3.1.2)

11. What is the effect of problematic data on downstream system performance? (3.1.3)

12. Can system behaviors and/or properties be accurately attributed to pretraining and/or fine-tuning data samples? (3.1.3)

### 3.1.1 Identification of Problematic Data

Motivation: Data plays a central role in the development and resulting capabilities of AI systems. Therefore, issues with data can propagate downstream, resulting in undesirable properties of models. We identify two ways in which data can be problematic.

The first is that data samples may violate some legal or ethical principle, simply by virtue of being included in a dataset. For instance, the presence of a sample could constitute a copyright or privacy violation (Brown et al., 2022; Rahman & Santacana, 2023; Subramani et al., 2023; Marcus & Southen, 2024), data poisoning (Biggio et al., 2012; Steinhardt et al., 2017; Wallace et al., 2020; Carlini & Terzis, 2021; Carlini, 2021; Schuster et al., 2021; Carlini et al., 2023a), or be inherently harmful (Thiel, 2023; Birhane et al., 2021; 2023; Luccioni & Viviano, 2021).

The second way that data could be problematic is if its use in training causes undesirable downstream effects. For instance, models trained on factually incorrect content in training data, such as vaccine disinformation, might replicate those factual inaccuracies. Indeed, (Lin et al., 2022b) demonstrate how models can "[generate] many false answers that mimic popular misconceptions." Alternatively, low-quality data, such as inaccuracies in low-resource languages, can compromise performance of models in those languages (Kreutzer et al., 2022).

Being able to identify problematic data used in the development of AI systems could facilitate various governance measures. For example, developers found to be using inherently harmful data could face penalties for improper use. Alternatively, developers found to be training on data that induces harmful model behaviors could face a higher burden of proof when demonstrating that their models are sufficiently safe for public deployment.

Open problems:

**Identifying problematic data given access to the training data.** For model developers with full access to the dataset, the major challenge is defining concrete, operationalizable criteria for detecting and removing problematic samples before training. Some problematic samples may be easier to identify than others; for instance, social security numbers can be identified with regexes or direct copies can be easily identified through pattern matching. However, the identification of other forms of problematic data samples poses more of a challenge. For example, understanding whether a data sample constitutes a copyright infringement requires knowledge of copyright law, making judgments about how much lexical similarity amounts to infringement, and the intended application of the data (Henderson et al., 2023a; Balganesh, 2012). Other approaches could resemble detection methods for data contamination, such as fuzzy string matching, or audio and video fingerprinting, as used by Youtube to identify copyrighted pieces (Cano et al., 2005; Wu et al., 2017).

**Identifying problematic data without access to the training data.** Regulators, auditors, or other entities who don't necessarily have access to a system's training data may need to find proxies for problematic data based on model behavior. Potential approaches include calculating confidence scores for the inclusion of data points (Li et al., 2024a), or using data watermarks introduced by creators that can be detected without access to the training corpora (Wei et al., 2024). Other approaches may be used to identify the use of copyrighted data, including inference attacks (Shokri et al., 2017) and influence functions (Grosse et al.,

2023; Choe et al., 2024). Yet, these approaches lack robustness – an issue that further research could aim to address.

**Tracing data provenance.** It can be challenging for model developers to ensure training data is correctly licensed due to licenses frequently being aggregated and misrepresented (Longpre et al., 2023). Hence, better tooling for data provenance will be necessary if developers are to feel comfortable that they are honoring creators' licenses. The Data Provenance Initiative (Longpre et al., 2023) conducted a large-scale audit of open-source fine-tuning data collections and cataloged corresponding data sources, licenses, and creators in an attempt to establish the provenance of data. However, the Data Provenance Initiative is limited in the datasets they cover, and expanding it to other data is resource-intensive. Automated license collection or standardized meta-data reporting for datasets (Longpre et al., 2024b) could help developers to release systems without facing legal ambiguity.

**Silent removal of harmful data.** Another challenge is that of being able to remove harmful data without the unwanted side-effect of publicly identifying it, and thus allowing malicious actors an easy way to source such data (Thiel, 2023). For example, the LAION-5B image dataset (Schuhmann et al., 2022) is composed of image URLs and metadata, as opposed to the images themselves. If one were to simply remove the URLs of the identified harmful images from the dataset, then this could provide malicious actors with the locations of these harmful images, either by comparing the datasets before and after removal, or directly through a repository change log where the dataset is hosted. However, care should be taken as methods for addressing this challenge could potentially also allow for the subversion of existing techniques (such as open change logs) for facilitating transparency into developers' data handling practices.

### 3.1.2 Infrastructure and Metadata to Analyze Large Datasets

Motivation: In addition to improved methods for auditing and identifying problematic samples in datasets, methods and infrastructure are needed for implementing these methods at scale. Contemporary datasets commonly contain on the order of tens of terabytes of data,[9] introducing the challenge of having the computational resources to store and handle such large quantities of data. Addressing this challenge will be crucial for enabling the identification of problematic data in practice.

Open problems:

**Automating meta-data collection for datasets.** A challenge with large-scale dataset infrastructure is that metadata – including links to the original data sources or license information – is not always provided (Piktus et al., 2023). An open research question is how the collection of metadata for previously published datasets could be automated. Additionally, exploring the automatic addition of cryptographic check sums as part of the metadata at the dataset level could help to enable users to confirm that the data they download matches the original data. Such an approach could be a partial solution to ensuring that datasets have not been altered (either maliciously or incidentally), especially in light of advances in data poisoning attacks (Carlini et al., 2023a).

**Determining relevant targets for dataset analysis.** Another question concerns the appropriate metrics for ascertaining the suitability of datasets for use in training, based on diffuse and macro-scale properties. For example, a dataset may be biased not as a result of the inclusion of individual samples, but the overall distribution of all included samples. Determining which measures and metrics are relevant in large-scale dataset analyses is an open challenge (Cho & Lee, 2021). A further open question concerns the information necessary for applying these metrics when evaluating datasets, extending the work on model and data cards by Mitchell et al. (2022b) and Pushkarna et al. (2022).

**Developing search tools for large datasets.** While there exists some work to quantitatively analyze dataset attributes (Mitchell et al., 2022b), this methodology generally "does not adapt well to web-scale corpora" (Piktus et al., 2023). One initial attempt to close this gap and to provide infrastructure for the quantitative and qualitative analysis of large datasets is the ROOTS Search Tool (Piktus et al., 2023). However, at the moment, ROOTS is limited to the 1.6TB corpus used to train BLOOM, and does not support other large-scale datasets. Extending this tool or creating similar tools for other open-access datasets could

---

[9]See for example, FineWeb (Penedo et al., 2024), a dataset of English text totaling 45 terabytes.

help with large-scale data governance. A related effort by Elazar et al. (2024) has provided an example of such an extension.

### 3.1.3 Attribution of Model Behavior to Data

Motivation: As noted above, one way in which training data may be problematic is if it causes undesirable downstream effects in models trained on that data, such as reproducing false information. In order to identify samples that should be excluded from training datasets for this reason, it may be necessary to understand how dataset composition can affect model performance. Thus, a complementary open issue to identifying problematic data is attributing model behavior to specific data points. Insights could be applied to different datasets used during development. For example, while pretraining data forms the basis for overall model performance (see e.g. Blakeney et al., 2024), preference data used during fine-tuning has a larger impact on the extent to which the resulting model is aligned with users' interests.

Open problems:

**Understanding how pretraining data affects model behavior.** Due to the size and complexity of contemporary AI systems, collective understanding of how specific training samples can contribute to problematic model behavior is incomplete (Lin et al., 2022a; Siddiqui et al., 2022). Udandarao et al. (2024) have, for example, investigated how pretraining concept frequency impacts downstream performance. Others have studied how upsampling domain-specific data relative to generic web-scraped text affects model performance (Blakeney et al., 2024).

**Understanding properties and effect of preference data on fine-tuned models.** In addition to pretraining data, future work could aim to understand the effects of the preference data used to fine-tune models, whether this is via a reward model, constitution, or another representation of preferences. It may be relevant to ensure that preference data has certain properties, such as representativeness, diversity, or neutrality, and design evaluations for the quality of this data. Related work in this context is research on preference aggregation for fine-tuning language models (Mostafazadeh Davani et al., 2022; Siththaranjan et al., 2024; Barnett et al., 2023) and scaling laws for reward model over-optimization (for example, Gao et al., 2023a).

**Understanding the impact of synthetic data.** Model developers may face data scarcity as current AI systems require increasing amounts of training data (Villalobos et al., 2022). Synthetic data generation offers an alternative to creating new authentic data, which can be both time-consuming and resource intensive. However, the impacts of training on synthetic data on model behavior are not yet well understood (Guo et al., 2023; Alemohammad et al., 2023; Martínez et al., 2023; Shumailov et al., 2023; Gerstgrasser et al., 2024). Questions remain on the effect that different strategies of training on synthetic data can have on model performance and bias, given that synthetic data has been shown to lack representativeness and insufficiently reflect imperfections of real-world data (Hao et al., 2024). Future research could aim to provide greater clarity regarding the effect of training on synthetic data, or develop uses of synthetic data for promoting model safety and reducing bias.

**Balancing tractability and accuracy in data attribution.** One key challenge in data attribution is the trade-off between computational tractability and accuracy (see, for example, Ghorbani & Zou, 2019; Jia et al., 2019; Ilyas et al., 2022; Akyürek et al., 2022), such as when using influence functions to attribute behaviors to data examples (Basu et al., 2020; Grosse et al., 2023; Choe et al., 2024). Park et al. (2023b) aim to overcome this trade-off by introducing TRAK (*Tracing with the Randomly-projected After Kernel*) but so far this work has only been tested for small foundation models. Due to TRAK's methodology requiring the training of multiple model versions for different subsets of the training set, it is unlikely that such a method would scale to the largest models, creating an avenue for future work.

## 3.2 Compute

---

**Example Research Questions**

---

13. What hardware properties or chip specifications are most indicative of suitability for AI training and/or inference? How does this differ from other scientific, business, or casual uses of high-end hardware? (3.2.1)

14. How efficiently can AI models be trained using a large number of small compute clusters? (3.2.1)

15. How can decentralized training attempts be identified? (3.2.1)

16. Can large training runs be detected while retaining developer privacy, for example through identifying signatures in processor utilization? (3.2.2)

17. Can compute workloads be reliably classified as either training, inference, or non-AI-related, for example through identifying signatures in processor utilization? (3.2.2)

---

### 3.2.1 Definition of Chip and Cluster Specifications for Model Training

Motivation: Compute governance has been proposed as a lever for governing advanced AI systems, due to the large amounts of computing resources required for their training and deployment (Sastry et al., 2024). However, despite its potential efficacy, compute governance is a blunt instrument, with previous actions including the restriction of the sale of a broad range of high-end chips to China (Bureau of Industry and Security, 2022b; 2024). It would thus be beneficial to be able to limit compute governance interventions to only the chips or compute clusters that are most relevant for developing and deploying AI systems of interest to policymakers. Ensuring that compute governance is targeted only where needed will require thoughtful derivation of metrics and specifications that capture hardware of concern, while excluding the vast majority of computing resources that are not used for industrial-scale AI.

Open Problems:

**Assessing the effect of chip specifications on AI workload suitability.** One open issue is understanding how different chip specifications – such as throughput, memory bandwidth, memory capacity, and interconnect bandwidth – affect a chip's suitability for different AI workloads, or the suitability of a cluster made up of those chips. Previous regulations concerning chips have arguably contained loopholes due to limitations in their technical specifications. For example, NVIDIA's A800, "while compliant with October 7 controls, is still capable of performing complex AI tasks, and was a top seller in China before the updated October 17, 2023 controls." (Reinsch et al., 2023)

**Understanding the implications of decentralized training and cluster size.** A related sub problem is understanding decentralized training, specifically the question of how efficiently AI models can be trained using multiple geographically disparate compute clusters. There is substantial technical work on developing decentralized training methods (see, for example, Douillard et al., 2024). Another open problem is the efficiency and cost impact of using a larger number of less powerful chips within a cluster, as opposed to using a smaller number of more powerful chips totaling the same theoretical throughput, sometimes known as *slicing.*

### 3.2.2 Classification of Workloads

Motivation: In addition to classifying hardware, it is also useful to classify computational workloads in order to identify potentially concerning or anomalous workloads. For example, Executive Order 14110 requires the reporting of training runs above a particular compute threshold (The White House, 2023a). Being able to classify workloads while preserving customer privacy could assist with such reporting, as well as a range of other governance goals, such as identifying compute usage trends, and audit trails for the development of powerful models (Heim et al., 2024; Shavit, 2023).

Open Problems:

**Privacy-preserving workload classification.** Compute providers, such as data center operators and cloud computing firms, typically already collect a wide variety of high-level data on customers and workloads (Heim et al., 2024). An open question is thus whether it is possible to use this data to develop reliable workload classification techniques, for example, determining whether a training workload exceeds certain compute thresholds, or whether an inference workload involves malicious cyberactivity (Commerce Department, 2024). Such techniques would need to account for changes in the hardware, software packages, and specific algorithms used in AI workloads over time.

**Ensuring workload classification techniques are robust to adversarial gaming**. Adversarial compute customers may try to obfuscate their activities to avoid workload classification, by introducing noise in the way computational resources are used, or breaking up workloads across multiple cloud accounts, providers, or computing clusters. Designing workload classification approaches that are robust to this kind of gaming, or otherwise are able to detect when this kind of gaming is occurring, is an open challenge (Egan & Heim, 2023; Heim et al., 2024).

### 3.3 Models and Algorithms

---

**Example Research Questions**

18. How can the thoroughness of evaluations be measured? (3.3.1)

19. How can potential blind spots of evaluations be identified? (3.3.1)

20. How can potential data contamination be accounted for when conducting evaluations? (3.3.1)

21. How can mechanistic analysis of model internals, such as weights, activations and loss landscapes on particular data, be used to improve understanding of a model's capabilities, limitations and weaknesses? (3.3.1)

22. How generalizable are mechanistic analyses across models? (3.3.1)

23. How can methods for red-teaming models be scaled and/or automated? (3.3.2)

24. How can the capabilities and risks of AI agents be evaluated? (3.3.3)

25. How can the capabilities and risks of networks of multiple interacting AI agents be evaluated? (3.3.3)

---

#### 3.3.1 Reliable Evaluations

Motivation: A great deal of research is focused on evaluating AI models to measure their performance and identify capabilities and failure modes, including by state actors (Department for Science, Innovation & Technology & AI Safety Institute, 2024; UK AI Safety Institute, 2024; Anthropic, 2024b). Yet, state-of-the-art AI systems can exhibit unpredicted downstream capabilities that often evade evaluations (Shayegani et al., 2023; Carlini et al., 2023b; Schaeffer et al., 2024). The rapid advancement and widespread deployment of AI systems has prompted major regions, including the European Union, China, and the United States, to put forward requirements for evaluating, reporting, and mitigating risks associated with these systems (Reuel et al., 2024a). For instance, Article 55 of the EU AI Act mandates that providers of general-purpose AI models with systemic risk perform model evaluations using standardized protocols and tools, including adversarial testing, to identify and mitigate systemic risks (Council of the European Union, 2024). Despite jurisdictions mandating capability evaluations across various risk areas, there is a lack of technical clarity on how to perform these assessments comprehensively and reliably (Chang et al., 2024; Zhou et al., 2023a), and for some risks such evaluations simply do not yet exist (Weidinger et al., 2023). In addition, evaluations for decision-making systems in high-stakes settings will likely demand a higher level of confidence than other applications, but it is unclear how to determine the required level of rigor based on use case.

Open Problems:

**Ensuring sufficient testing.** Determining whether an evaluation procedure has identified all, if not most, of the vulnerabilities of a system is an open problem. This is especially relevant if the evaluation pertains to capabilities, such as deception and long-horizon planning, that could enable harmful forms of misuse or make systems hard to oversee or control (Park et al., 2024; Shevlane et al., 2023; Hendrycks et al., 2023; Kinniment et al., 2023; Li et al., 2024b; Phuong et al., 2024; Bengio et al., 2024). This issue is most acute in the case behavioral evaluations that directly test models' performance on benchmarks or test cases (Liang et al., 2023; Srivastava et al., 2023; Gao et al., 2023b; Wang et al., 2023a; Lee et al., 2023b; Biderman et al., 2024), as such evaluations can fail to fully inform evaluators about a model's capabilities, with particular uncertainty around capabilities that the model may lack (Casper et al., 2024a). Specifically, if a model does not exhibit a particular behavior during testing, it is challenging to determine whether this is due to the model genuinely lacking the underlying ability or if the evaluation method applied was insufficient to surface it (Wei et al., 2022; Zhu et al., 2023a), for example, due to prompt sensitivity (Zhu et al., 2023a; Sclar et al., 2023). This ambiguity can lead to an incomplete understanding of a model's true capabilities and potential risks (Barrett et al., 2023; Schaeffer et al., 2023; Raji et al., 2021), and means that behavioral evaluations can only offer a lower bound on a model's potential to exhibit harmful behavior (Goel et al., 2024; Casper et al., 2024b). This motivates research into how to expand the scope of behavioral evaluations as well as how to estimate the robustness and extensiveness of existing evaluations (Chan, 2024).[10]

**Improving evaluations using mechanistic analysis.** The shortcomings of behavioral evaluation techniques motivate additional work on evaluation methods that do not suffer from the same limitations. One proposed approach toward this is to study their internal mechanisms (Burns et al., 2022; Olah, 2023; Carranza et al., 2023; Casper et al., 2024a; Bereska & Gavves, 2024). Although evaluations that involve developing interpretations of a model's internal structure have seen a great deal of interest, they remain largely untested in practice (although see Templeton et al., 2024; Gao et al., 2024). Furthermore, interpreting the internal mechanisms of models inevitably involves reductions in the complexity of the system being studied (Gilpin et al., 2018).

**Testing the validity of evaluations.** It can be hard to have high confidence that the results of the evaluations reflect properties of the model rather than the evaluation methodology employed – that is, that the evaluation is internally valid.[11] Making progress on this could be challenging due to uncertainties pertaining to both evaluations and the models to which they are applied, making it difficult to attribute results to either the evaluation or the model. One potential avenue for progress could be using *model organisms* – smaller simpler AI models that have particular properties by construction (for example Hubinger et al., 2024) – to test whether evaluations for the constructed property are able to reliably detect it. However, this method would likely not be of use if trying to develop evaluations for model properties that are only exhibited by the most capable models. More generally, designing meta-evaluations that assess the reliability and consistency of evaluation methodologies across different models and contexts remains an open research challenge.

**Establishing a causal relationship between procedural design choices and system characteristics.** Similar to attributing a model's properties to characteristics of its training data (see Section 3.1.3), it may be possible to attribute behavior or performance to design decisions made during a model's development process (see, for example, Simson et al., 2024). Establishing causal relationships between such decisions and resulting system properties may allow for greater standardization of development best practices.

**Understanding potential risks and capabilities of future AI systems.** Finally, it is also difficult to study certain risk scenarios that might emerge through advanced capabilities – for example, the ability to plan over a long horizon – due to their hypothetical nature. It could be useful to develop and study specific demonstrations of harmful behavior and the effectiveness of current safety techniques against these behaviors

---

[10]Beyond insufficient model-level testing, evaluations along the whole AI life cycle are lacking. We address this problem separately in Section 3.4.

[11]Internal validity refers to the extent to which an evaluation accurately measures what it intends to measure within its specific context, while external validity refers to how well the evaluation results can be generalized or applied to other situations, populations, or contexts beyond the original study.

in current AI systems, similar to the model organisms approach suggested in the previous paragraph (Scheurer et al., 2024; Järviniemi & Hubinger, 2024; Hubinger et al., 2024).

### 3.3.2 Efficient Evaluations

Motivation: Ideally, it would be possible to test an AI system under all possible inputs to ensure that it would not produce a harmful output for any of them. However, performing a brute-force search over possible inputs is intractable due to the astronomically large input spaces for modern systems,[12] and the fact that whether an output is harmful may be unclear or context-dependent. As a result, current evaluation methods manually apply heuristics to guide vulnerability searches towards regions of the input space assumed to be more concerning – as observed in voluntary audits by developers (OpenAI et al., 2024; Kinniment et al., 2023; Touvron et al., 2023; Anthropic, 2023b). However, manual attacks quickly become impractical, expensive, and insufficient for conducting scalable evaluations (Ganguli et al., 2022), especially when searching across modalities and languages (Üstün et al., 2024). This problem is exacerbated for increasingly capable, general-purpose systems that have a significantly larger attack surface than narrower systems. The development of more efficient automated approaches for identifying model vulnerabilities will be crucial if results from model evaluations are to be applied as inputs to governance-relevant decisions.

Open Problems:

**Making comprehensive red-teaming less resource-intensive.** Recent progress has been made on automated red-teaming by using generative AI models to produce test cases (Perez et al., 2023; Shah et al., 2023), develop adversarial prompts (Deng et al., 2022; Perez et al., 2022; Mehrabi et al., 2023; Hubinger et al., 2024; Casper et al., 2023; Hong et al., 2024), and automatically evaluate the outputs of other models (Zheng et al., 2023; Chiang et al., 2023; Ye et al., 2023; Kim et al., 2023; Chao et al., 2024; Souly et al., 2024). Furthermore, automated search methods have been applied to find adversarial attacks as a way of enhancing or replacing manual methods (Wallace et al., 2019; Song et al., 2020; Shin et al., 2020; Guo et al., 2021; Shi et al., 2022; Kumar et al., 2022; Wen et al., 2023; Jones et al., 2023; Zou et al., 2023b; Liu et al., 2023b; Zhu et al., 2023b; Andriushchenko et al., 2024). However, despite these approaches for automating evaluations, thorough red-teaming remains a labor-intensive process, and many failure modes still evade red-teaming efforts (Shayegani et al., 2023; Carlini et al., 2023b; Longpre et al., 2024a). These challenges in part stem from how existing automated techniques are often computationally expensive and crude, requiring a large degree of guidance from human engineers (Mazeika et al., 2024). Qualitatively different approaches that automate some or all the red-teaming process – perhaps through the use of agentic AI systems that can plan, use tools, and dynamically evaluate systems – could also allow for more scalable evaluation.

### 3.3.3 (Multi-)Agent Evaluations

Motivation: *Agentic* AI systems are generally characterized by an ability to accomplish tasks from high-level specifications, directly influence the world, take goal-directed actions, and perform long-term planning (Chan et al., 2023b; Durante et al., 2024; Huang et al., 2024a). These capabilities could allow agentic systems to perform tasks with little human involvement and control. While economically useful, for example, as customized, personal assistants, or for autonomously managing complex supply chains, agentic systems could pose unique risks due to their ability to directly act in the world, potentially with difficult-to-predict impacts (Chan et al., 2023b; Lazar, 2024; Gabriel et al., 2024; Bengio et al., 2024).

Open Problems:

**Evaluating and monitoring agentic systems.** User customizability, such as through prompting or the integration of new tools, makes it particularly difficult to foresee the use cases and potential risks of agents (Shavit et al., 2023; Kolt, 2024; Cohen et al., 2024a), motivating potential measures for tracking and monitoring their actions (Chan et al., 2024). Furthermore, evaluating agents is a nascent field with significant challenges – existing agent benchmarks often don't have adequate holdout datasets, causing existing agents to game and overfit to the benchmark, which in turn results in unreliable evaluations of these systems

---

[12]For example, there are vastly more possible 20-token strings of text or $10 \times 10$ pixel images than there are particles in the observable universe. (One current estimate for the number of particles in the observable universe is $10^{80}$. GPT-2's tokenizer had a vocabulary size of $50,257$, meaning that there are approximately $50,000^{20} \approx 10^{94}$ unique strings of 20 tokens. There are $256^{100} \approx 10^{240}$ possible $10 \times 10$ gray-scale images with integer pixel values in the range $[0, 255]$.)

(Kapoor et al., 2024b). Similar to non-agent benchmarks (see above), best practices are currently lacking, leading to inconsistencies across evaluations and limiting their reproducibility (Kapoor et al., 2024b). Thus, future work could aim to introduce best practices for evaluating agentic systems.

**Expanding limited multi-agent evaluations.** On top of the difficulties of studying single-agent systems, multi-agent interactions add an additional layer of complexity due to information asymmetries, destabilizing dynamics, and difficulties in forming trust and establishing security. These problems can lead to unique complexity and failure modes (Hammond et al., 2025; Chan et al., 2023a; Akata et al., 2023; Mukobi et al., 2023). In addition, it may be the case that collectives of agents exhibit unpredictable capabilities or goals not attributable to any one agent in isolation (Hammond et al., 2025). If AI agents become increasingly embedded in real-world services, such as in finance or the use of web services, it will be relevant to understand such multi-agent dynamics.

**Attributing downstream impact to individual agents.** For issues of liability, it will be critical to be able to determine which agent(s) or system(s) can be held responsible for a particular decision or action, if any. This may be complicated for cases in which the cause is not solely attributable to a single AI agent.[13] Having methods of tracing multi-agent interactions and determining the cause of a particular outcome could help to solve this problem. An open technical question in this context regards techniques for monitoring individual agents' contributions to multi-agent systems, in order to ease attribution of responsibility (Friedenberg & Halpern, 2019).

### 3.4 Deployment

---

**Example Research Questions**

26. How can the downstream societal impacts of AI systems be predicted and/or determined? (3.4.1)

27. How can downstream impact evaluations be scaled across languages and modalities? (3.4.1)

28. How can benchmarks be designed in a way that ensures construct validity and/or ecological validity? (3.4.1)

29. How can dynamic simulation environments be designed to better reflect real-world environments? (3.4.1)

---

#### 3.4.1 Downstream Impact Evaluations

Motivation: The performance of models in isolation is an imperfect proxy for the impact that AI systems will have in everyday use. Thus, comprehensively understanding the impacts that AI could have on society demands robust methods for evaluating systems in dynamic, real-world settings (Ibrahim et al., 2024). Having such methods would allow policymakers a higher-fidelity picture of where governance intervention might be necessary in order to address potential harms.

Open Problems:

**Predicting the downstream societal impacts of AI systems.** Although understanding overall societal impact is an overarching goal of much work on evaluating AI, it is a difficult, large-scale sociotechnical problem (Dolata et al., 2022; Solaiman et al., 2023; Rakova & Dobbe, 2023; Weidinger et al., 2023; Dobbe & Wolters, 2024; Bengio et al., 2024). For example, simplified technical proxies for complex concepts like *fairness* and *equity* insufficiently measure the disparate impact of AI systems on diverse communities (Blodgett et al., 2020; Selbst, 2021). It may also be logistically challenging to conduct downstream evaluations. For example, while it is known that large language models exhibit significant cross-lingual differences in safety and capabilities (Yong et al., 2023; Wang et al., 2023c;b; Jin et al., 2023; Üstün et al., 2024), it is expensive and time-consuming to thoroughly evaluate the cross-lingual properties of models due to the required coordination between speakers of many languages. Ultimately, thoroughly assessing downstream societal impacts

---

[13]See, for example, (Wex Definitions Team, 2023)

requires nuanced analysis, interdisciplinarity, and inclusion (Hagerty & Rubinov, 2019; Bengio et al., 2024). While recent work has taxonomized societal impacts and provided an overview of early techniques for their evaluation (Moss et al., 2021; Shelby et al., 2022; Raghavan, 2023; Solaiman et al., 2023; Weidinger et al., 2023; 2024), there remains a lack of structured, effective methods to quantify and analyze these impacts.

**Ensuring construct validity of evaluations.** It can be difficult to establish confidence that the proxy used in an evaluation or benchmark accurately captures the concept it aims to measure – that is, its construct validity (Raji et al., 2021; Bowman & Dahl, 2021; Hutchinson et al., 2022; Subramonian et al., 2023; McIntosh et al., 2024). For example, while MMLU (Hendrycks et al., 2021) claims to assess a model's understanding and memorization of knowledge from pretraining through the proxy of performance on question answering, it is unclear how well the ability to accurately answer questions serves as an indicator for understanding. Future research could aim to evaluate the construct validity of current benchmarks as well as ensure construct validity for AI system evaluations, perhaps taking inspiration from prior work in psychology (see, for example, (Westen & Rosenthal, 2003; Strauss & Smith, 2009; Smith, 2005)).

**Ensuring ecological validity of evaluations.** In addition to construct validity, establishing the ecological validity of benchmarks is an open challenge.[14] Current benchmarks tend to be biased towards easily-quantifiable and model-only metrics, potentially making them ill-suited for predicting how well models perform when deployed in real-world settings (Ouyang et al., 2023; Lee et al., 2023a). Future work could aim to assess the correlation between performance on benchmarks and downstream performance, as well as propose evaluation methods for which this correlation is tighter. Another consideration is that benchmarks used for assessing capabilities are oftentimes used to guide development of models, limiting the extent to which such benchmarks can be seen as unbiased (Marie et al., 2021; Dehghani et al., 2021; Madaan et al., 2024; Salaudeen & Hardt, 2024), and making it challenging to know how to interpret resulting scores.

**Designing dynamic evaluations and real-world simulation environments.** Despite many user interactions with AI systems taking place in dynamic, multi-turn environments, current benchmarks only evaluate performance in such settings to a limited degree (Ibrahim et al., 2024). By creating dynamic evaluation frameworks (for example, Dynabench, 2023; Park et al., 2023a), researchers could better assess a system's performance, inherent characteristics, and potential risks in a more realistic manner compared to static test sets. This would require significant investment in infrastructure and tooling, such as sophisticated simulated environments tailored to specific domains like hacking, persuasion, or biosecurity. Similarly, experiments with human subjects are valuable to understand the risks from human capability increases through AI models (Mouton et al., 2023), but are less common due to their resource-intensiveness.

---

[14] *Ecological validity* refers to the extent to which results from experiments generalize to contexts outside of the testing environment. Compare this to *construct validity* which refers to the extent to which an assessment measures the target construct.

# 4 Access

Many governance actions will likely require third-party access to system components, along with the provision of resources such as compute. As examples, external access will likely be a necessary consideration for facilitating third-party audits (Raji et al., 2022b; Anderljung et al., 2023b; U.S. National Telecommunications and Information Administration, 2024; Mökander et al., 2023; Casper et al., 2024a), evaluations by government AI safety institutes (GOV.UK, 2023; National Institute of Standards and Technology (NIST), 2024; Department for Science, Innovation and Technology et al., 2023b), and independent academic research (Bucknall & Trager, 2023; Kapoor et al., 2024a; Longpre et al., 2024a; House of Commons Science, Innovation and Technology Committee, 2024).

There are numerous reasons for why it may be desirable to enable these functions to be performed by those other than the system developers. Firstly, actions such as evaluation and auditing could benefit from the independence of being conducted by parties external to AI developers, so that developers do not "mark [grade] their own homework" (Gerken & Rahman-Jones, 2023). Furthermore, AI developers may not have the capacity or incentives to conduct research to the extent needed for advancing our scientific understanding of AI systems at a rate comparable to that of advances in AI development, motivating the need for involvement of the broader academic community. Third-party access may also allow for broader cultural diversity and representation in the development and governance of AI (Dobbe et al., 2020; Delgado et al., 2023; Held et al., 2023; Crowell, 2023), though should not be seen as a sufficient measure (Chan et al., 2021; Sloane et al., 2022).

However, many of the aforementioned actions are precluded due to insufficient external access to relevant system components, especially in the case of the most state-of-the-art systems (Solaiman, 2023; Bommasani et al., 2023a; Bucknall & Trager, 2023; Casper et al., 2024a). This is often due to concerns regarding developers' intellectual property, privacy of data subjects, legal uncertainty, and the safety of the system in question (Seger et al., 2023).

Access to compute and other resources peripheral to the systems under consideration, such as training data, is also essential for many auditing and research functions (Ahmed & Wahed, 2020; Besiroglu et al., 2024; Ojewale et al., 2024). The challenges associated with facilitating access to such resources will also need to be addressed if these functions are to be fulfilled in academia and the public sector.

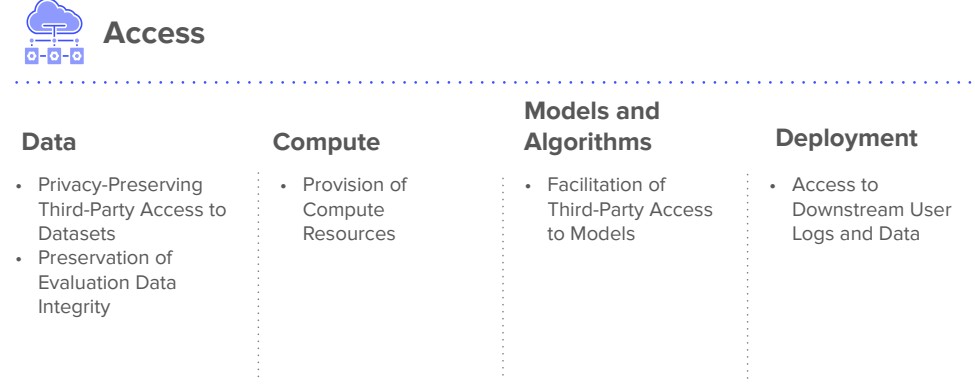

Figure 3: Open problem areas in the *Access* capacity, organized by target

## 4.1 Data

---

**Example Research Questions**

30. How can data access be structured so as to preserve privacy while enabling meaningful auditing? (4.1.1)

31. How can data access be reconciled with privacy-preserving machine learning? (4.1.1)

32. How can openly hosted datasets be prevented from contaminating training data? (4.1.2)

33. How can independent evaluation on standardized datasets be facilitated without openly hosting evaluation datasets? (4.1.2)

---

### 4.1.1 Privacy-Preserving Third-Party Access to Datasets

Motivation: Access to training datasets is crucial for enabling external data audits that aim to identify instances of harmful, personal, or inappropriate data being included in datasets (Thiel, 2023; Birhane et al., 2021; 2023; Subramani et al., 2023; Luccioni & Viviano, 2021, see also Sections 3.1 and 5.1). External access to datasets could also be instrumentally necessary for facilitating assessment and research into models (Bucknall & Trager, 2023; Ojewale et al., 2024), for example, as research into how the content of training datasets influence downstream model behavior depends on visibility into the training datasets (for example Udandarao et al., 2024, see also Section 3.1).

However, naïvely providing unrestricted access to datasets may violate legal bounds, for example by reproducing copyrighted data, or otherwise raise security concerns. Additional challenges stem from the lack of legal clarity regarding how to define problematic data, such as what constitutes a copyright violation in the context of AI training (Henderson et al., 2023a; Quang, 2021). Furthermore, developers may be reluctant to provide unrestricted access to proprietary datasets due to the high costs associated with collating such datasets,[15] the risk of leaking sensitive intellectual property, or, in some cases, risking disclosing having knowingly trained on illegally collected data. These concerns may be able to be alleviated by providing greater transparency into the makeup, contents, and aggregate characteristics of datasets, as well as information about sources and compilation methodologies, in the absence of unrestricted access to the entire dataset.

Open problems:

**Structuring external access to datasets to reduce privacy risks.** Research could aim to develop methods for providing sufficiently deep access to datasets, for example, for the purpose of auditing and evaluation, while protecting privacy of data subjects. Existing work in this direction includes Trask et al. (2023) which suggests a framework for allowing third-parties to propose and execute approved queries on AI systems and third-party data without exposing sensitive information beyond the explicitly authorized results. Similarly, *Project Oak* is a software package for providing "security and [...] transparency around how data is used and by whom, even in a highly distributed system", through the use of trusted execution environments (TEEs) on specialized hardware (Project Oak, see also Section 6.2). Inspiration can also be taken from other industries, including healthcare (NHS Research SDE Network, 2024). For example, *OpenSAFELY* provides a platform for the analysis of patient healthcare data for the purposes of academic research through a combination of pseudonymization, methods for working with data *in situ* while obfuscating raw patient data, and providing transparency into researchers' use of the platform (Bennett Institute for Applied Data Science, 2024). Future research could draw insight from this case to propose methods for safely accessing user data for research into the societal impacts of AI.

**Addressing the tension between data access and privacy-preserving machine learning.** A significant body of research has explored how we can train machine learning models when data-owners do not trust

---

[15]Including the cost of data collection and the resources to ensure that licenses are correctly represented to avoid incorrect infringements.

model-trainers – for example through federated learning (Kairouz et al., 2019) and training on encrypted data (Xie et al., 2014; Nandakumar et al., 2019). In such cases, it can be challenging to provide data access given that the developer themselves lacks such visibility. Future work could explicitly address this tension, proposing methods that allow for the auditing of data that may have been encrypted during training.

### 4.1.2 Preservation of Evaluation Data Integrity

Motivation: Current standardized methods for evaluating models often utilize openly-available datasets, with the goal of comparing model performance like-for-like (Reuel et al., *forthcoming*). At present, this is largely achieved by hosting such datasets openly in online repositories such as *HuggingFace*[16] (see, for example, Hendrycks et al., 2021; Srivastava et al., 2023; Gao et al., 2023b). However, openly hosting evaluation datasets online risks their inclusion in web-scraped training datasets (Deng et al., 2023), either accidentally, or intentionally as a method for artificially inflating benchmarking results. Such contamination of training data has serious implications for the efficacy and reliability of these standardized metrics (Oren et al., 2023; Roberts et al., 2023; Jiang et al., 2024; Zhang et al., 2024; Schaeffer, 2023)

Open problems:

**Identifying and mitigating contamination of training datasets.** Current approaches to mitigating contamination of training datasets are rudimentary. A potential post-contamination approach is to detect data contamination (Dong et al., 2024b; Golchin & Surdeanu, 2023) and then correct for it when scoring benchmarks. For example, OpenAI and Meta measured which benchmarks' test samples were potentially included in the pretraining data of GPT-4 and Llama 2, respectively, and reported how scores differed between contaminated test samples and non-contaminated test samples (OpenAI et al., 2024; Touvron et al., 2023). Zhang et al. (2024) attempted to correct for data contamination by creating a version of the benchmark GSM8k that is comparable in terms of tasks, complexity and human solve rates, and reported which models' scores dropped precipitously. Other approaches pertain to the design of benchmarks that are robust against contaminated models by using templates from which variations of a task can be generated (Yu et al., 2023; Srivastava et al., 2024) or by being frequently updated (White et al., 2024). However, the frequent updating approach is resource intensive, especially if covering a large span of test tasks and fields, and designing templates to support variations of tasks may not be feasible for tasks that don't follow a predictable structure. Alternatively, (Srivastava et al., 2023) make use of a *canary string*, that is, a globally unique identifier that is included in all sub repositories of the BIG-bench collection in order to ease identification of these test samples in training datasets. BIG-bench also includes a dedicated `training_on_test_set` task, which serves as a "post-hoc diagnosis of whether BIG-bench data was used in model training" (Srivastava et al., 2023). However, use of canary strings depends on any and all copies of the repositories to also include the string, and thus is not robust to negligent users.

**Evaluating on private or encrypted evaluation datasets.** Alternatively, research could aim to develop ways in which models can be independently evaluated on a private or encrypted test set. Indeed, some recent benchmarking datasets have only been made available only through a custom evaluation API (see, for example, Sawada et al., 2023). Furthermore, popular dataset repositories including *HuggingFace*, as well as competition platforms such as *Kaggle*, gate access to evaluation datasets in order to reduce the risk of contaminating training data.[17] Finally, Bricman (2023) proposes *hashmarks*, a "protocol for evaluating language models in the open without having to disclose the correct answers" by cryptographically hashing a benchmark's reference solutions before publication. Further work could develop this, and similar protocols for reliably evaluating system capabilities on private evaluation data.

---

[16]https://huggingface.co/
[17](Hugging Face, 2024; kaggle, 2024)

## 4.2 Compute

---

**Example Research Questions**

34. How can public compute resources be allocated fairly and equitably between users? (4.2.1)

35. How can public compute infrastructure be developed in a way that ensures interoperability between models and software packages? (4.2.1)

36. How can assurance be given that researcher compute provisions are being used for intended and stated purposes? (4.2.1)

---

### 4.2.1 Addressing Compute Inequities

Motivation: Compute usage by private companies in training and running models has increased exponentially in the past years, and now greatly exceeds the compute resources available for non-industry researchers (Maslej et al., 2024; Besiroglu et al., 2024). While some researchers have found that the majority of academic researchers do not feel primarily constrained by compute access (Musser et al., 2023), others have found that this access inequality is specifically limiting researchers' contribution to frontier research (Ahmed & Wahed, 2020; Besiroglu et al., 2024; Birhane et al., 2023). To address these concerns, there have been proposals and funding for public compute infrastructure (NDIF proposal; Ho et al., 2021; Organisation for Economic Co-Operation and Development, 2023; National Artificial Intelligence Research Resource Task Force, 2023; UK Research and Innovation, 2023). Though the success of these initiatives primarily depends on raising funds to purchase sufficient compute, technical advances could still be instrumental in their success.

Open problems:

**Ensuring interoperability of public compute resources.** Public compute resources should be compatible and interoperable with a wide range of models and software packages, in order to support the range of research projects that would be conducted. System performance can vary considerably depending on the hardware and software on which it is run (Nelaturu et al., 2023; Gundersen et al., 2022), and common ML software frameworks can lose more than 40% of their key functionality when ported to non-native hardware (Mince et al., 2023). Future research could aim to propose solutions that address these observed defects.

**Ensuring environmental sustainability of public compute resources.** The resourcing requirements of large-scale supercomputers and data centers are considerable, both in terms of energy (Strubell et al., 2019) and other resources such as water, used for cooling (Mytton, 2021). Thus, measures will need to be taken to balance broad access to computing resources with environmental sustainability. The environmental impacts of AI systems and associated open problems is discussed further in Section 8.

**Ensuring public compute is used for intended purposes.** System administrators would need to be able to ensure that public compute resources are being used for the stated purposes, rather than malicious or otherwise unintended uses, for example by performing "workload classification" (Heim et al., 2024, see Section 3.2 for open problems in this context). Such oversight methods would need to preserve end-users' privacy to system administrators, as well as that of any potential subjects of data used in conducting experiments (see Section 5.2).

**Equitably allocating public compute resources.** Given the high demand for public compute resources, another issue will be the efficient and fair allocation of processor time between users. While methods for allocating compute resources in more general cases have been explored (Ghodsi et al., 2011; Wang et al., 2015; Xu & Yu, 2014; Souravlas & Katsavounis, 2019; Jebalia et al., 2018), future work could aim to ensure their applicability to this case. Alternatively, research could aim to find AI-specific optimizations for allocating compute among diverse users.

### 4.3 Models and Algorithms

> **Example Research Questions**
>
> ---
>
> 37. What research and auditing methodologies are possible given a range of forms of access on the continuum between black- and white-box access? (4.3.1)
>
> 38. How do different forms of access affect potential risks of misuse of models? (4.3.1)
>
> 39. How do different forms of access on the continuum between black- and white-box access affect the risk of model theft or duplication? (4.3.1)
>
> 40. How can model access requirements for research and auditing be reconciled with commercial and/or safety concerns? (4.3.1)

#### 4.3.1 Facilitation of Third-Party Access to Models

Motivation: A fundamental requirement of conducting external research and evaluation of AI systems is having access to the underlying models. However, many systems are not released openly (Solaiman, 2023), and, while access requirements vary widely for different external actors, current APIs do not offer sufficient depth or flexibility of access to facilitate many actions important for research and evaluation (Bucknall & Trager, 2023; Casper et al., 2024a; Longpre et al., 2024a). For example, Casper et al. (2024a) argue that evaluations conducted with solely black-box access can "produce misleading results" and only offer "limited insights to help address failures" due to their not revealing complete information regarding the nature of discovered flaws. It is a challenge to find the balance between providing external parties with sufficient access for conducting independent research and evaluation, while addressing developers' concerns such as IP theft or misuse of their models. While there are certainly social and legal pathways that can be pursued towards this end, there are also several technical avenues (Casper et al., 2024a).

Open problems:

**Illuminating the continuum between black- and white-box access.** Greater clarity regarding how much, and what kinds of, research can be conducted with different depth and breadth of access would be helpful for navigating the trade off between access and security (Bucknall & Trager, 2023). Different auditing procedures can demand varied levels of access, motivating the need for a range of methods for supporting researchers, rather than prescribing a set approach (Casper et al., 2024a). On the flip side, a clearer picture of how differing forms of access bear on developers' security and privacy concerns would also be needed. Black-box access already allows for the training of *distilled models* that can then be used to generate effective adversarial attacks against production models via transfer (Zou et al., 2023b), and fine-tuning APIs can be ineffective at guarding against the removal of pre-deployment safety measures (Qi et al., 2023). Meanwhile, the ability to view language model output logits has been shown to be sufficient for extracting proprietary system information, including the model's hidden dimension, though it is unclear the extent to which this is a practical threat (Carlini et al., 2024). Further research could aim to elucidate how the provision of intermediate forms of grey-box access could exacerbate these existing vulnerabilities when compared to the baseline of black-box access.

**Applying technical measures to address vulnerabilities of greater access provisions.** Outstanding technical questions regarding external model access include whether the use of technical tools, such as privacy-enhancing technologies or TEEs, could enable near-white-box auditing and research while addressing commercial and safety concerns. For example, Aarne et al. (2024) describe how an approach combining multi-party computing with TEEs "could be used by a third-party evaluator to run tests on an AI model without ever having direct access to the unencrypted weights." Future research could assess the extent to which such solutions allow model providers, auditors, and regulators to interact in a way that is: easy to set up for the model provider; leaks no model information to the auditor; leaks no audit information to the model provider; and does not compromise the cybersecurity of the model provider. Alternatively, work could aim to incorporate approaches for providing third-party model access at scale into secure and trusted compute

clusters, including public compute resources (Anderljung et al., 2022; Heim, 2024). To date, there has been preliminary work on protocols for secure privileged evaluations (Trask et al., 2023), though there is a lack of existing applications or established best practices.

**Ensuring version stability and backward-compatibility of hosted models.** Large commercial models are frequently and continually updated during deployment, with prior versions often being replaced without notice or knowledge. However, reproducibility and replicability of independent research conducted on proprietary models depends on stable and continued access to models, even after their being succeeded by newer versions (Pozzobon et al., 2023; Bucknall & Trager, 2023; Biderman et al., 2024). Maintaining hardware and software compatibility (Mince et al., 2023) may be necessary in order to be able to provide access to discontinued systems upon request. Future work could lay out best practices for documenting and communicating when models are being deprecated or discontinued, in order to ease reproducibility concerns.[18]

### 4.4 Deployment

---

**Example Research Questions**

41. How can user logs and data be used for downstream impact assessments while preserving the privacy of data subjects? (4.4.1)

42. How could responsibilities for providing user data access be effectively allocated along the AI value chain? (4.4.1)

43. What cryptographic methods can be developed to allow analysis of user interaction data without revealing individual user identities or sensitive information? (4.4.1)

44. How could secure multi-party computation be leveraged to allow collaborative analysis of user logs across different entities in the AI value chain? (4.4.1)

---

#### 4.4.1 Access to Downstream User Logs and Data

Motivation: Assessing models post-deployment, as covered in Section 3.4, requires access to relevant real-world data on user interactions with systems (Nicholas, 2024). This data can be used to directly assess aspects of user-model interactions, build evaluations that are more reflective of real-world use, and guide assessments of societal-level patterns in key sectors (Ibrahim et al., 2024). While there are crowd-sourcing initiatives which allow users to voluntarily submit some of their interaction data to create research datasets (The Allen Institute for Artificial Intelligence, 2024; ShareGPT, 2022), to the best of our knowledge, no model provider has made their interaction datasets, or privacy-preserving metadata about these logs, widely available.

Access to real-world user data, such as usage logs and user feedback records, may be additionally relevant for legal purposes. For example, legal cases may arise when users experience harm as a result of an interaction with an AI system – either directly as a participant in the interaction of concern, or indirectly as a subject of actions taken following an interaction. In such cases, the availability of information including user logs and audit trails to prosecutors or courts may be relevant for determining the outcome of a case.

Open problems:

**Addressing user privacy concerns regarding access to user logs.** External access to user interaction data must overcome privacy concerns relating to the collection, sharing, and analysis of potentially sensitive and identifying user information. This challenge has parallels in other industries, for example in online platform governance, where the EU's Digital Services Act (Digital Services Act) mandates independent researcher access to platform data (EU Joint Research Centre, 2023; Albert, 2022). While implementation challenges remain in the case of the Digital Services Act (Leerssen, 2021; Leerssen et al., 2023; Leerssen, 2023; Jaursch et al., 2024; Morten et al., 2024), more developed solutions can be seen in the healthcare

---

[18]See (Luccioni et al., 2022) for related work on deprecating datasets.

sector (see the discussion of OpenSAFELY in Section 4.1). Inspiration could be taken from these sectors for ensuring user privacy while providing access to user logs for research purposes.

**Understanding how access responsibilities may vary along the AI value chain.** Additional difficulties in providing user data stem from the complexities of the AI value chain (Küspert et al., 2023) – for example, in the case that a foundation model is built upon and incorporated in a user-facing application by a downstream deployer. In this scenario, it is not immediately clear how access requirements interact with the division of information between the foundation model developer and subsequent deployer. An additional challenge may emerge if the provision of user data in this case implicitly reveals information about how the deployer is using the model as part of their service, potentially putting their IP at risk of leakage. Further work could clarify potential access responsibilities in this and similar situations.

# 5 Verification

In many cases, it may be beneficial to be able to verify claims regarding AI systems' properties, capabilities, and safety as a way of increasing trust between actors (Brundage et al., 2020). While related to assessment, covered in Section 3, verification concerns the process of checking whether an AI system "complies with a [specific] regulation, requirement, specification, or imposed condition" (IEEE, 2011), as opposed to evaluating the system's performance, capabilities, or potential societal impacts. For example, an assessment task could be to uncover details about the data that a given model was trained on. In contrast, a verification problem could be, given a dataset and a model, to confirm or refute the claim that the model was trained on the dataset.

There is a trend towards an increasing amount of regulation and corresponding requirements for model developers, deployers, and users being passed by major national and international jurisdictions (Maslej et al., 2024). It may thus be necessary for model developers and deployers to verify and attest to certain properties of their AI systems in order to prove that they comply with regulations. On the flip side, governments may need to be able to verify whether actors in the AI ecosystem comply with the regulations, or verify that other countries are in compliance with international rules (Baker, 2023; Shavit, 2023; Avenhaus et al., 2006).

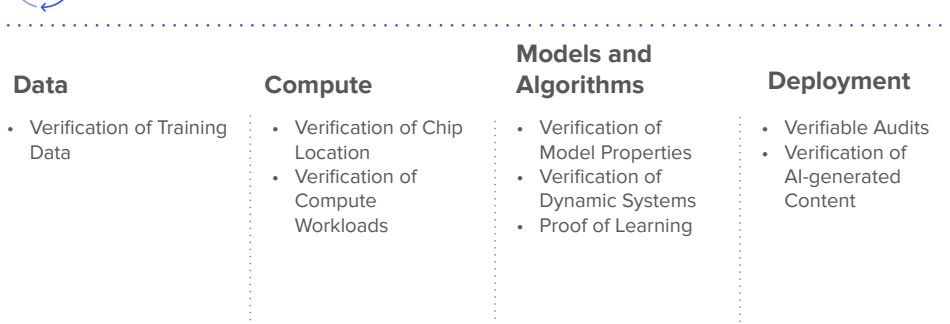

Figure 4: Open problem areas in the *Verification* capacity, organized by target

## 5.1 Data

---

**Example Research Questions**

45. How can it be verified that a model was (not) trained on a given dataset? (5.1.1)

46. How can it be verified that a dataset has certain properties, or (does not) include certain information? (5.1.1)

47. How can membership inference attacks be optimized for large-scale verification of training data in black-box settings? (5.1.1)

48. How could the verification process for correct use of licensed data in AI model training be formalized? (5.1.1)

---

### 5.1.1 Verification of Training Data

Motivation: Being able to verify the data on which a given model was trained – either as a model developer or third-party auditor – could aid in demonstrating compliance with data handling standards and regulation, including the Blueprint for an AI Bill of Right, or the EU's General Data Protection Regulation (The White

House Office of Science and Technology Policy, 2023; European Commission, 2016). In particular, even if it is possible to demonstrate that a dataset does not contain harmful or copyrighted material (see Section 3.1), this is insufficient for guaranteeing that a model was not trained on problematic data, as the developer may have used an alternate dataset to the one assessed. Being able to *post hoc* provide evidence that a model was trained on a specific dataset would help to preclude such instances.

Open Problems:

**Verifying datasets used to train a model.** Choi et al. (2023), building on Shavit (2023), formalize the *proof-of-training-data problem*, wherein a prover aims to "prove to a verifier that the resulting target model weights $W^*$ are the result of training on data $D^*$," and propose a solution. However, the authors concede that their approach is not robust to all potential attacks, in particular additions of small amounts of harmful data, such as that used for inserting backdoors (Xu et al., 2021). In addition, Choi et al.'s protocol is not applicable to models for which data is not fully known before training, as in the case of online or reinforcement learning. Finally, their protocol requires that the "Prover disclose confidential information to the Verifier, including training data, model weights, and code" (Choi et al., 2023), which may create IP and security concerns. An alternate approach to verifying training data could be the application of *membership inference attacks* which aim to infer whether a given data point was contained in a given model's training data in a black-box setting (Shokri et al., 2017; Duan et al., 2024a; Wei et al., 2024). Future research could aim to suggest robust methods for verifying training data, assess the robustness of existing methods, or address the parallel issue of verifying that a given dataset (or sample) *was not* used in the training of a given model.

**Verifying fair data use.** Verifying copyright compliance and fair data use is a complex challenge that may require additional legal and technical frameworks (see Section 3.1) to extend or replace the Proof-of-Training-Data approach (Choi et al., 2023). Open challenges include formalizing the verification of correct use of licensed data, as well as verifying the exclusion of specific licensed data from training sets, should the respective license not allow the data to be used for training a model.

## 5.2 Compute

---

**Example Research Questions**

49. How can the location of AI hardware be verified? (5.2.1)

50. How can on-chip geolocation mechanisms be made robust to existing GPS spoofing methods? (5.2.1)

51. Can TEEs be used to robustly attest to the identity of the specific chip, or the data that it is processing? (5.2.2)

52. What methods can be used to verify compute usage without the use of TEEs? (5.2.2)

53. How can TEEs and their applications be designed in a way that limits their potential for misuse, for example through unnecessarily-broad surveillance? (5.2.2)

54. How can the computational overhead of verification mechanisms be reduced to a level that enables application across large compute clusters? (5.2.2)

---

### 5.2.1 Verification of Chip Location

Motivation: High-end data center AI chips are the subject of U.S. export controls, but are at present straightforward to smuggle (Huang, 2024; Harithas, 2024; Fist & Grunewald, 2023). One key technical problem for enforcement is that it's not currently possible to know the location or owner of a chip after it has been exported. Being able to verify a chip's location could also help users of cloud computing validate that their (or their customers') data is being processed in accordance with local data processing laws.

Open Problems:

**Verifying chip location.** Accurately detecting a chip's location remains an open challenge. One approach could be to measure verifiable latencies between the chip in question and a network of trusted servers (Aarne et al., 2024; Brass & Aarne, 2024). Other methods could be valuable for verifying that a large number of chips are co-located in a single data center, and have not been resold or dispersed. It may be possible to verify this using a proof of work challenge that would require a large number of co-located systems to complete in the allotted time (Jakobsson & Juels, 1999). Alternatively, chips could more directly verify their identities to each other through mutual attestation (IETF Datatracker, 2024). It is worth noting that non-technical solutions for verifying chip location, involving the physical inspection of data centers, have also been proposed (Shavit, 2023; Baker, 2023).

**Designing hard-to-spoof chip IDs.** Secure IDs for chips (either software- or hardware-based) could help with traceability and preventing unauthorized use. For example, unique identifiers and information may be engraved during routine water fabrication flows (MULTIBEAM, 2024), or physical unclonable functions – devices that "[exploit] inherent randomness introduced during manufacturing to give a physical entity a unique 'fingerprint' or trust anchor" (Gao et al., 2020) – may be added. Open questions include understanding the security of these approaches, as well as their feasibility, given potential trade offs between usability, security, and effects on chip performance.

### 5.2.2 Verification of Compute Workloads

Motivation: It could be valuable for AI developers and deployers to be able to reliably verify their compute usage, for example, which chips were used to train their models and for how long. Likewise, chip owners, such as cloud providers, may want to demonstrate which models were trained on their compute, so as to provide evidence that their compute is not being used for unreported large-scale training runs.[19] Such mechanisms may be of particular utility in international and extra-territorial situations in which levels of trust between verifier and prover may be limited. Any verification scheme would need to uphold privacy of both user data and intellectual property.

We note that this is one area of research that itself is dual-use. While potentially beneficial in the above example cases, some implementations of such mechanisms could enable far-reaching control and monitoring of AI chips. However, careful design of these mechanisms can limit the scope of powers given to a regulator, for example by only requiring data flows for voluntary verification, rather than remote monitoring or control.

Open Problems:

**Verifying properties of workloads using TEEs**. It may be possible to use TEEs (or other hardware security technologies) to attest to the exact program code and model being run (Chen et al., 2019, see also Section 6.2). For example, TEEs could, given inputs along with a function to evaluate on them, return a signature that accompanies the output from the computation attesting to the computation being run as intended. Alternatively, the chip could also return a hash of the inputs and outputs of the computation. This would allow the prover to keep the inputs and outputs private, instead proving ownership by demonstrating that they can generate the hash returned by the chip during computation. A verifier could also use a TEE to confidentially run arbitrary tests on the model weights or other data. Despite promising theoretical possibilities, further work will be required to be able to implement the above in practice. In particular, firmware to implement the above solution on existing hardware is lacking, and morover, may need to be unusually secure. Advancements will also be needed if solutions such as the above are to be feasible at the largest scales, without introducing prohibitive overhead costs.

**Verifying properties of workloads with a trusted neutral cluster.** If TEEs are unavailable or impractical, another approach could be to save hashed snapshots of neural network weights during training along with information about the training run being conducted – that is, a *training transcript*. This information may then be used to verify that the training transcript provided would have resulted in the weights, with the use of a trusted neutral cluster (Shavit, 2023). Current challenges to implementing this procedure include

---

[19]Reporting of training runs above the threshold of $10^{26}$ floating-point operations (FLOP) is required, for example, by the US Executive Order on Safe, Secure, and Trustworthy Development and Use of Artificial Intelligence (Executive Office of the President, 2023).

difficulties in accounting for randomness in training procedures, building sufficiently trustworthy neutral clusters, and finding efficient methods for proving the authenticity of training transcripts that scale to the largest models (Shavit, 2023).

**Verifying compute usage of large, non-AI workloads.** Owners or users of large clusters may wish to demonstrate that their clusters were used for a large, non-AI workload (for example, climate simulations), as such use would not fall under the purview of AI regulation. One approach is workload classification, discussed above. There may also be viable computational approaches to verification, potentially including analogues to proof-of-learning methods (Jia et al., 2021), that could be explored in future research.

## 5.3 Models and Algorithms

---

**Example Research Questions**

55. How can model properties be verified with full access to the model? (5.3.1)

56. How can the risk associated with a given context, query, and AI response be assessed in order to obtain assurances about the system's compliance with safety requirements? (5.3.1)

57. What should constitute the lower bar for tracking updates to models, for example in a model registry? (5.3.2)

58. Could proof-of-learning be used to demonstrate and verify model ownership? (5.3.3)

59. How can proof-of-learning mechanisms be made robust to adversarial spoofing? (5.3.3)

---

### 5.3.1 Verification of Model Properties

Motivation: In order for system developers or deployers to demonstrate compliance with regulatory requirements, it may be necessary to prove claims regarding model properties and information. Verifiable properties could include model architecture, training procedures, or performance metrics, enabling developers to formally demonstrate compliance with any mandated technical specifications.

Open Problems:

**Verifying claimed capabilities and performance characteristics with full model access.** Model properties could be verified through formal verification methods if the verifier has full access to the model, as in the case of the model developer. Such methods aim to mathematically prove that a given system can(not) respond in particular ways to particular inputs (Katz et al., 2017b;a; Kuper et al., 2018; Katz et al., 2019). For instance, formal verification was used to study the safety of neural networks used for unmanned aircraft collision avoidance (Irfan et al., 2020). However, such methods remain largely untested for advanced AI models (Dalrymple et al., 2024). In particular, many methods quickly become prohibitively complex when scaled up to contemporary state-of-the-art models. While there exist additional methods for verifying properties such as performance metrics without full access (see Section 5.4.1), research could focus on more efficient methods given full access to the model. Furthermore, verifying properties such as a system's architecture or training procedure remain open questions.

### 5.3.2 Verification of Dynamic Systems

Motivation: Modern AI systems, such as ChatGPT, are not based on static models. Rather, they consist of multiple models and components, for example, mixture-of-experts, input filters, and output filters, that undergo change throughout their life cycle. This poses an oversight challenge due the ever-changing nature of many systems throughout their deployment life cycle. Having a reliable, accessible process for versioning could help to monitor system updates and their impacts.

Open Problems:

**Tracking versioning and updates.** Key open questions in this context relate to how model versioning and post-deployment modifications should be kept track of, especially for models that undergo frequent updates. One approach could be to have registries that track models over time, however, it's not clear what information should be stored in such a registry, nor how the information could be verified. Other approaches that can be useful as a starting point to verify dynamic models include reward reports for reinforcement learning (Gilbert et al., 2023), ecosystem graphs (Bommasani et al., 2023c), or instructional fingerprinting of foundation models (Xu et al., 2024).

### 5.3.3 Proof-of-Learning

Motivation: In the current landscape, there is no mechanism for a model developer to prove that they have invested the computational resources required to train a given model. Such a proof could be used for resolving ownership disputes when models are released or stolen by allowing the developer to attest to their having trained the model (Tramèr et al., 2016; Orekondy et al., 2018; Jia et al., 2021). Additionally, proof-of-learning could aid in defending against accidental or malicious corruption of the training process when performing distributed training across multiple workers (Li et al., 2014; Jia et al., 2021).

Open Problems:

**Scalable proof-of-learning.** (Jia et al., 2021) were the first to formalize the notion of proof-of-learning for AI models. The authors demonstrated that stochastic gradient descent accumulates "secret information due to its stochasticity," which they show can be used to construct "a proof-of-learning which demonstrates that a party has expended the compute required to obtain a set of model parameters correctly" (Jia et al., 2021). Alternatively, (Goldwasser et al., 2021) develop *Probably Approximately Correct* verification, in which a weak verifier interacts with a strong prover to test whether the model trained by the prover has a low loss relative to the best possible model, with respect to a given loss function. Scaling these techniques such that they remain practical given the growing training compute budgets of foundation models is an open challenge.

**Designing adversarially robust proof-of-learning.** Since the introduction of proof-of-learning in (Jia et al., 2021), subsequent work has demonstrated its vulnerability to adversarial attacks – that is, false proofs that are cheap for an adversarial prover to generate (Zhang et al., 2022; Fang et al., 2023). In particular, Fang et al. (2023) demonstrate "systemic vulnerabilities of proof-of-learning" and which depend on advances in understanding optimization to be sufficiently addressed. While Choi et al. (2023) suggest a protocol to counter these vulnerabilities through memorization-based tests, and fixing the initialization and data order, they only test their protocol for single attacks and not for composite attacks. Their protocol further only covers language models, and has not been tested for other modalities. Future work could aim to assess these claims, and aim to increase the robustness of proof-of-learning to adversaries.

## 5.4 Deployment

| **Example Research Questions** |
| --- |
| 60. How can audit registries be used to provide end-to-end verification along the AI value chain? (5.4.1) |
| 61. How should verification information from model registries be presented to users? (5.4.1) |
| 62. Can zero-knowledge proofs be applied to demonstrate a model's compliance with hypothetical mandated criteria, without directly disclosing architectural details? (5.4.1) |
| 63. How can it be verified that the model version on which an evaluation or audit was performed is the same as is deployed? (5.4.1) |
| 64. How can the implementation of safety measures be verified at deployment? (5.4.1) |
| 65. How can output watermarking schemes be made robust to adversarial attempts at removal? (5.4.2) |

66. How can metadata watermarking be applied to AI-generated content? (5.4.2)

67. How robust can AI content detectors be expected to be in light of continuing advances in generative AI? (5.4.2)

68. How should AI-generated content detectors handle cases of genuine images that have been modified or edited with AI tools? (5.4.2)

### 5.4.1 Verifiable Audits

Motivation: As discussed in Section 3, external audits and assessment have been proposed as crucial components of governance regimes (Raji et al., 2022b; Mökander et al., 2023). Being able to attest to an audit's process and outcome could establish greater trust between model developers, third-party auditors, and governments by proving compliance with regulatory requirements. Trust could also be established with end-users by enabling them to verify that the model with which they are interacting has been shown to have the properties claimed by developers (Godinot et al., 2024), for example in model cards (Mitchell et al., 2019), official communications (for example, Anthropic, 2024a), or technical papers (Gemini Team et al., 2023; OpenAI et al., 2024). Verifying audit results is often made more challenging as a result of access to models often being restricted due to IP and security concerns (South et al., 2024).

Open Problems:

**Verifying claimed capabilities and performance characteristics without full model access.** Preliminary work has explored how the application of zero-knowledge proofs to AI systems can enable privacy-preserving verifications of claimed system properties, as well as confirmation that model weights used for inference match those on which an audit was run (South et al., 2024; Waiwitlikhit et al., 2024; Sun et al., 2024; Yadav et al., 2024). However, due to the high computational overhead associated with these methods, addressing speed constraints will be necessary if such methods are to be applied to larger models. Current approaches that future work could build on include GPU acceleration (Sun et al., 2024) or proof splitting (South et al., 2024).

**Verifying audit results at inference time.** In theory, verified computing — such as through TEEs (Sabt et al., 2015), zero-knowledge proofs (Fiege et al., 1987), or secure multi-party computation (Goldreich, 1998) with active security — could facilitate verifiable audits in a two-stage process. In the first step, a model developer could load an inference pipeline[20] into a cluster of enclave computers. Upon an auditor concluding their study of the system (for example, based on the approaches outlined in the previous paragraph and in 5.3.1), they could ask the cluster of enclaves to produce a certificate of the pipeline that was evaluated, which is then stored in a public audit registry. In the second stage, a user, when interacting with the secured pipeline, could request a corresponding certificate with each received generation. Using such a method, this consumer could know that the AI pipeline they're receiving generations from is the same pipeline that was previously evaluated to be safe. If any change was made to the pipeline, the certificates would not match, and the user would know that they're receiving generations from a pipeline which has not been evaluated. However, given the dynamic nature of current models in use (see Section 5.3.2), changes may occur more frequently than audits of such models, which poses an open challenge to this proposal. Another open problem is that this pipeline requires that all evaluation and inference is done in enclaves and with significant computational overhead, effectively limiting verifiable audits for a few critical systems, and necessitating more scalable structures for verifying audits. It will also be necessary to find a way for consumers to be informed of the outputs of this verification in a low-friction way, as in the case of browsers that provide warnings for websites without HTTPS certificates. Finally, it is unclear how secure this method is against attempts to exfiltrate model weights by auditors.

**Verifying use of safety measures post-deployment.** In safety-critical settings, regulators may want to ensure that safety measures, for example, output filters, are applied to AI models or their outputs (see, for example, Dong et al., 2024a; Leslie et al., 2024; Welbl et al., 2021). Enforcing this may require methods for auditing systems deployed in such domains to check that they do in fact have safeguards that meet

---

[20]Typically consisting of input pre-processing, model prediction, and post-processing of the model's output.

these specifications. An open question is how to enforce that additional filters, classifiers, modifications are attached to models deployed in safety-critical domains.

### 5.4.2 Verification of AI-generated Content

Motivation: The ability to distinguish between AI-generated and authentic content may be instrumental in verifying the authenticity of information and maintaining public trust in information ecosystems. Stipulations for being able to detect AI-generated media are made in several regulatory efforts, for example in Article 50 of the AI Act (Council of the European Union, 2024) which are, given the state of the art of detection and verification tools, currently unrealizable (Zhang et al., 2023). Methods for verifying AI-generated content can roughly be divided into *ex ante* approaches that mark AI-generated content as such by embedding machine-readable watermarks, and *ex post* methods that aim to classify content as either AI-generated or not, in the absence of a watermark (Ghosal et al., 2023). In addition, watermarks could potentially be used to verify that AI-generated content was created by a particular model, improving accountability by facilitating the identification of responsible parties in case of unintended consequences or misuse.

Open Problems:

**Developing robust watermarking schemes.** Watermarks – signals placed in output content that are imperceptible to humans, but easily detectable through application of a specific *detector* algorithm – have been proposed as one method for verifying that a particular model generated a given output (Kirchenbauer et al., 2023; Christ et al., 2023; Saberi et al., 2023). However, the level of robustness of watermarks varies between modalities. In particular, the continuous output space of images and audio enables hidden watermarking that is more effective than for text (Ghosal et al., 2023). As such, future work could aim to address the relative lack of robustness of watermarks in the case of AI-generated text (Zhang et al., 2023; Liu et al., 2023a). Additionally, research could aim to address the possibility that watermarks are easy to fake – for example, having two similar models that produce watermarks that cannot be distinguished (Srinivasan, 2024).

**Designing robust AI content detectors.** While efforts to develop methods for the detection of AI-generated content have seen increased attention in the last two years (Sadasivan et al., 2023; Corvi et al., 2023; Berber Sardinha, 2024), these methods have not always held up to independent evaluation (Weber-Wulff et al., 2023). As generative systems improve, it will be increasingly difficult to develop methods (machine learning-based or otherwise) to distinguish their output from genuine media. Continued work will need to be done in order to improve and maintain the efficacy of AI content detectors in light of this continued advancement.

**Utilizing verifiable meta-data to identify authentic content.** An alternative to identifying AI-generated content could be to develop ways for a content creator to verify their content as AI-generated or authentic by adding verifiable meta-data to it (Jain et al., 2023c; Knott et al., 2023). For example, the Coalition for Content Provenance and Authenticity (C2PA) is tackling this issue by developing standards for the certification of the provenance of digital content (C2PA, 2022). Similar work is being done by the (Content Authenticity Initiative, 2024). This could be useful for AI labs to label content, but also for creators of non-AI-generated media to label their authentic content as such using the same standard. However, a significant drawback of this approach is that it is not robust to adversaries, as meta-data can easily be stripped from the content – a limitation which future research could aim to address.

**Verifying authentic content modified using AI.** Complications arise when going beyond the binary distinction of AI-generated content on the one hand, and human-generated on the other. For example, it is currently unclear how AI content detectors should respond to authentic images that have been modified using generative AI tools. Future work could aim to assess the suitability of AI-content generation tools for detecting such cases, or design detectors that are able to distinguish between AI-generated, AI-modified, and authentic content.

# 6 Security

In this section, we consider security in the context of AI governance, which aims to ensure that unauthorized actors are not able to access systems and infrastructure not intended for their use, nor use systems for malicious purposes. Being able to give security guarantees across system components could be helpful for a number of reasons. Increased security can strengthen a wide array of governance actions through reducing the risk of regulatory requirements being subverted. For example, comprehensive security measures can protect the confidentiality of training data, ensuring that AI systems developed using sensitive personal information remain in compliance with data protection laws.

It should be noted that security is one of the areas of this report that comes closest to topics within AI safety. As such, many of the topics discussed below under the umbrella of TAIG could also be viewed through the framing of improving AI safety, or otherwise be closely related to topics that can. However, due to the reasons above, we decided to include security within this report nonetheless.

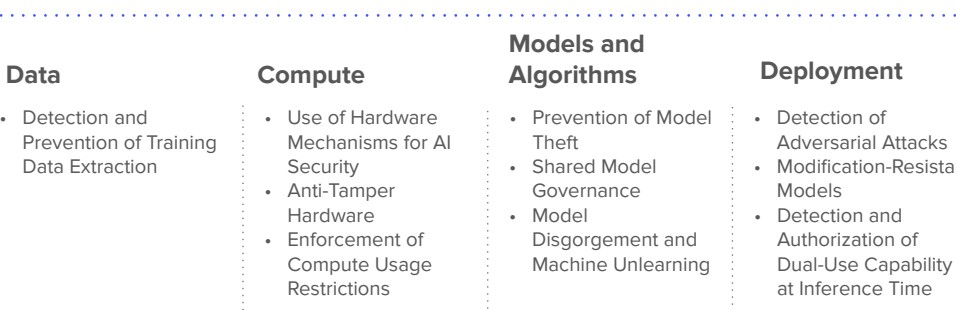

Figure 5: Open problem areas in the *Security* capacity, organized by target

## 6.1 Data

| Example Research Questions |
| --- |
| 69. How can attempted data extraction attacks be reliably identified? (6.1.1) |
| 70. How can AI systems be made robust to data extraction attacks? (6.1.1) |
| 71. How can methods for restricting *verbatim* reproduction of training data be generalized to protect the same information being extracted in a slightly different form? (6.1.1) |

### 6.1.1 Detection and Prevention of Training Data Extraction

Motivation: Prior research has demonstrated how large amounts of models' training data can be extracted *verbatim*, with a variety of methods applicable in both black- and white-box settings (Carlini et al., 2023c; Nasr et al., 2023; Shi et al., 2023; Balle et al., 2022; Carlini et al., 2019; 2021; Duan et al., 2024b; Prashanth et al., 2024). Short of building models that are robust to extraction attacks, having the ability to detect them could enable API-level defenses that can block model outputs upon detection of a potential attack, or the introduction and enforcement of litigation against perpetrators of extraction attacks.

Open Problems:

**Improving system robustness to extraction attacks.** De-duplication of training data has been shown to assist in reducing memorization, and hence extraction, of specific data-points (Kandpal et al., 2022), though

Nasr et al. (2023) suggest that this provides only marginal improvement. Alternatively, there may be *post-hoc* interventions that do not alter a model's memorization of its training data, but nonetheless decrease its propensity to reproduce it, with one such example being machine unlearning (see Section 6.3.3). However, acute challenges still remain. For example, restricting the *verbatim* reproduction of training samples does not prevent the same information being generated by models with slight rewording or reformatting (Ippolito et al., 2023). Furthermore, guarding against the reproduction of specific samples has been found to expose previously-safe samples to the same attacks – a phenomenon dubbed the "Privacy Onion Effect" (Carlini et al., 2022). Finally, a related yet under-explored area of concern regards the potential for large-scale data extraction from retrieval datasets (Qi et al., 2024c).

**Detecting attempted data extraction attacks.** Proposed methods for detecting potential data extraction attacks are noticeably absent from the literature, with most publications on this topic aiming to identify model vulnerabilities to such attacks. Potential methods for detecting extraction attacks could focus on either the model inputs or outputs, allowing model providers to filter out suspicious prompts, or outputs that bear a close resemblance to training samples, respectively.

## 6.2 Compute

---

**Example Research Questions**

72. How can hardware-enabled governance methods be implemented at scale to ensure the security of a compute cluster? (6.2.1)

73. How can it be ensured that given code, along with model weights, can only be executed with a license that is verified on-chip, so that distributed AI executables can only be run on approved chips? (6.2.1)

74. How can on-chip governance firmware be modified or updated while the chip is in operation while remaining resistant to potential attacks? (6.2.1)

75. How secure are existing implementations of TEEs on AI accelerators? (6.2.1)

76. How can methods for tamper-evidence or responsiveness be reconciled with the performance demands of high-end AI accelerators? (6.2.2)

77. How secure are existing approaches to tamper-evidence and responsiveness? (6.2.2)

78. How can tamper-proofing methods incorporate self-destruct mechanisms in case of attempted tampering? (6.2.2)

79. How could the use of high-end chips in training foundation models be prevented? (6.2.3)

80. Can enforceable mechanisms be developed to allow for the export of chips under predefined conditions? (6.2.3)

---

### 6.2.1 Use of Hardware Mechanisms for AI Security

Motivation: The integration of hardware mechanisms such as TEEs into AI computing clusters could ensure the confidentiality and integrity of workloads (Li et al., 2023b; Geppert et al., 2022; Mo et al., 2024) while also greatly aiding with AI security and attestation (Nevo et al., 2024; Kulp et al., 2024; Aarne et al., 2024). This in turn would assist in implementing many of the aforementioned problem areas relating to verification and access.

Open problems:

**Ensuring utility of TEEs for hardware-enabled governance and security.** While TEEs have seen broad adoption on CPUs,[21] application to AI accelerators (such as GPUs and TPUs) has so far been limited. The most notable example is Nvidia's incorporation of a TEE in its H100 GPU, referring to it as "NVIDIA Confidential Computing" (Dhanuskodi et al., 2023; Hande, 2023; Apsey et al., 2023). However, the H100 implementation "still may not support all of the mechanisms required for an ideal implementation of [hardware governance measures]" (Aarne et al., 2024). In particular, questions remain regarding whether TEEs can robustly attest to the identity of a specific chip or the data that it is processing. Furthermore, current implementations mostly do not support confidential computing across multiple individual accelerators in a cluster. Further work could investigate the extent to which such functions are supported by current and next-generation chips, or aim to specify hardware, firmware, or software requirements necessary for robustly implementing such functions at the scale of compute clusters, or even entire data centers.

**Ensuring security of TEEs on AI accelerators.** Given the novelty of the application of TEEs to high-end, AI-specific hardware, it is as-yet unknown how secure such systems are in practice due to a lack of independent testing. Previous independent testing of CPU TEEs has uncovered numerous potential vulnerabilities (Muñoz et al., 2023; van Schaik et al., 2022). Additional security research into GPU TEEs and other security features that they rely on, such as Nvidia's GPU system processor, would be valuable for identifying areas of improvement and assessing whether these features can be relied on for different governance applications.

### 6.2.2 Anti-Tamper Hardware

Motivation: Some of the aforementioned hardware-enabled governance mechanisms, such as verifying compute workloads (Section 5.2), rely on the assumption that the hardware in question has not been compromised or tampered with, for example, if dependent on a TEE for implementation. However, well-resourced adversaries may aim to physically tamper with chips in order to obviate the defensive protections installed on them. It could therefore be useful to disincentivize or prevent such tampering. This can come in the form of *tamper evidence*, whereby physical manipulation of the chip is detectable after the fact, or through *tamper responsiveness*, whereby tampering with the chip triggers an automatic response ranging from deletion of sensitive information stored on the chip, known as *zeroization*, to permanent self-destruction.

Open problems:

**Reconciling tamper evidence and responsiveness with practical requirements of state-of-the-art AI hardware.** While approaches to tamper evidence and responsiveness have been proposed (Immler et al., 2018), ensuring that such approaches are compatible with the unique requirements of state-of-the-art AI accelerators, while retaining affordability and scalability, is an outstanding challenge (Aarne et al., 2024). For example, demanding cooling requirements and high-bandwidth interconnect between chips pose a challenge due to the need to bridge the interior and exterior of the tamper-proof enclosure (Obermaier & Immler, 2018).

**Ensuring the robustness of anti-tamper approaches.** Established approaches to tamper evidence and responsiveness have depended on the use of specialized packaging[22] that encases the chip and is unable to be removed without leaving visible traces of damage. In the case of tamper responsiveness, the packaging may carry an electric current such that damage disturbs the current, acting as a trigger for zeroization or other active responses. More advanced methods use physical unclonable functions – a method that "exploits inherent randomness introduced during manufacturing to give a physical entity a unique 'fingerprint' or trust anchor" (Gao et al., 2020) – to remotely attest that a chip has not been tampered with (Immler et al., 2018; 2019; Obermaier & Immler, 2018). However, evidence pertaining to the practical success of tamper-proofing has been limited (Aarne et al., 2024; Kulp et al., 2024). Further research is needed if we are to increase confidence in anti-tamper measures when applied to AI hardware security.

---

[21]See, for example, specifications by Intel (Intel, 2022) and AMD (AMD, 2024).

[22]In this context, *packaging* refers specifically to a physical security enclosure that encases a hardware device as opposed to packaging, such as a cardboard box or other container, in which the device may be stored or transported.

### 6.2.3  Enforcement of Compute Usage Restrictions

Motivation: Recent attention in compute governance has been paid to export controls placed on cutting-edge chips of the type used in large-scale training of AI systems (Bureau of Industry and Security, 2022a; Allen, 2022). However, export controls are a blunt instrument with a high potential for collateral damage by restricting the sale of affected chips for legitimate uses. Indeed, the Bureau of Industry and Security, the US executive agency responsible for such export controls, itself put a call out for "public comments on proposed technical solutions that limit items specified under [the export controls] from being used in conjunction with large numbers of other such items in ways that enable training large dual-use AI foundation models with capabilities of concern" (Bureau of Industry and Security, 2023), presumably acknowledging that technological developments for disentangling legitimate and malicious uses of high-end chips would be desirable. It is worth noting, however, that there is considerable disagreement over both the viability of such solutions, with concerns raised regarding the level of confidentiality of such measures, as well as the possibility of their being circumvented (Ting-Fang, 2023; Patel, 2023; Grunewald & Aird, 2023; Fist & Grunewald, 2023).

Open problems:

**Implementing remote attestation for disaggregated machines.** It would be useful to verify the particular set of hardware components – in this case, AI chips – that are part of the same cluster. This would assist with hardware-based methods for verifying properties of workloads, which typically rely on knowing which chips are participating in a workload, and for ensuring that end-users are complying with export control obligations. An open question is how remote attestation could work for disaggregated machines (Google Cloud, 2024) or for heterogeneous devices. These mechanisms could allow the nature and acceptability of the configuration to be remotely attested to. Other relevant projects in this context that may serve as starting points are *Caliptra* and *OpenTitan*.

**Restricting particular cluster configurations.** It may also be possible to restrict possible cluster configurations to assist with export control policies. One proposed approach involves restricting the communication bandwidth between GPUs to prevent "many consumer device-chips from being aggregated into a supercomputer" (Kulp et al., 2024). It may be possible to build such a system on top of existing features such as trusted platform modules (Hosseinzadeh et al., 2019), or it may require new protocols and new hardware-level features – all of which are open problems.

## 6.3  Models and Algorithms

---

**Example Research Questions**

81. What cybersecurity measures can be taken at the infrastructure level to protect model weights from theft by an adversary? (6.3.1)

82. How can models be protected from inference attacks aiming to reproduce or replicate model weights and architecture? (6.3.1)

83. What are the most promising methods for enabling shared model governance? (6.3.2)

84. How should the success of different model unlearning techniques be evaluated? (6.3.3)

85. How can it be ensured that machine unlearning and model editing techniques do not cause unwanted side-effects such as removing concepts that were not explicitly targeted? (6.3.3)

86. How effective are model unlearning and model editing techniques when applied to multi-lingual or multi-modal models? (6.3.3)

---

### 6.3.1 Prevention of Model Theft

Motivation: As models become more capable they could become an increasingly valuable target for theft by adversarial parties wanting to put them to their own potential (mis)use. Similarly, as state-of-the-art models become more broadly integrated into the economy and society, the attack surface will increase, potentially leading to a greater threat of exfiltration (Nevo et al., 2024). It follows that securing model weights, and other system components, might become an increasing priority to prevent theft or model access by unauthorized parties that may undermine governance initiatives aimed at ensuring customer safety and national security (Nevo et al., 2024).

Open problems:

**Ensuring adequate cybersecurity for model weights.** Protecting model weights against exfiltration attempts requires protections against insider and outsider threats (Nevo et al., 2024). This includes standards for physical security of the data center facility itself, as well as of the hardware and software stacks (OpenAI, 2024b).[23] Improved coordination between actors facing similar threats might also assist defenders in understanding the threat landscape and better protecting their assets during training and deployment. Further analysis of potential threat vectors, as well as development of physical and cybersecurity measures including and beyond those in (Nevo et al., 2024), would help to identify and address these risks.

**Defending against model inference attacks.** Alternatively, adversaries may try to extract or replicate models through attacks to a query API (Orekondy et al., 2018; Tramèr et al., 2016; Jagielski et al., 2020; Carlini et al., 2020; 2024), logit values (Carlini et al., 2024) or side-channel attacks (Wei et al., 2020). Further research could aim to quantify threats and develop methods for defending against these, and other, forms of model extraction attacks.

### 6.3.2 Shared Model Governance

Motivation: Shared model governance refers to the practice of distributing control over a model's training or inference across multiple parties, such that training or inference can only be carried out with the agreement of all parties (Bluemke et al., 2023). The ability to distribute control of a model in this way could have many potential use cases – for example, if multiple diverse actors want to pool investment for training a shared model where each actor has specific requirements for how the model is trained. This may also be applicable in the case of international collaboration between state-backed institutes wanting to collaborate on AI research (Ho et al., 2023).

Open problems:

**Enabling shared model governance through model splitting.** One proposed approach for technically-enforced shared model governance is *model splitting* (Martic et al., 2018) – that is, "distributing a deep learning model between multiple parties such that each party holds a disjoint subset of the model's parameters." Martic et al. (2018) also investigate the resulting question of how computationally expensive it would be for a single actor to reconstruct the entire model, starting from their share of the parameters. A similar approach is taken by *SplitNN* (Vepakomma et al., 2018; Ceballos et al., 2020), though the emphasis is placed on model splitting to achieve data privacy, rather than shared model governance. Given the relatively small amount of prior work on model splitting for shared model governance, future work could aim to provide further proofs of concept and evaluate their efficacy.

**Enabling shared model governance through secure multi-party computation and homomorphic encryption.** Two potential alternative approaches for achieving shared model governance are applying either secure multi-party computation (SMPC) (Yao, 1982; 1986; Evans et al., 2018), or homomorphic encryption (HE) (Gentry, 2009; Acar et al., 2018). Though usually applied to AI for the purposes of data privacy (Knott et al., 2021; Kumar et al., 2019; Tan et al., 2021; Guo et al., 2022; Riazi et al., 2018), or model privacy (Trask, 2017; Dahl, 2017; Ryffel et al., 2018), both SMPC and HE could potentially be leveraged for shared model governance. For example, using SMPC, a model creator could take each parameter within a model and split it into multiple *shares*, distributing such shares across *shareholders*. Alternatively, using HE, a model could be encrypted using one or more private keys such that the ability to decrypt model results relies upon

---

[23]Data center security standards: (International Organization for Standardization, 2021; Wikipedia contributors, 2023).

the application of the private key of all parties. However, both HE and SMPC have performance concerns, with even state-of-the-art encrypted deep learning approaches yielding 100x performance slowdown (Wagh et al., 2019; 2020; Stoian et al., 2023; Frery et al., 2023). Future work could investigate this potential in more detail, aiming to provide proof-of-concept demonstrations of how shared model governance using either HE or SMPC could be achieved. Alternatively, future research could aim to reduce the high computational overheads of HE and SMPC.

**Enabling shared model governance through TEEs.** Finally, shared governance could potentially be achieved through the application of TEEs. In this case, multiple parties could upload a program to a TEE, with the chip providing evidence on the software program being run. With this evidence, all parties can know how any information they upload to the enclave will be handled. TEEs may also be able to reinforce the above approaches, for example, SMPC. Future research could aim to provide proof-of-concept demonstrations of shared model governance through the use of TEEs given the current hypothetical nature of this approach.

### 6.3.3 Model Disgorgement and Machine Unlearning

Motivation: The concepts of *model disgorgement* (Achille et al., 2024) and *machine unlearning* (Bourtoule et al., 2021; Nguyen et al., 2022; Shaik et al., 2023; Si et al., 2023; Eldan & Russinovich, 2023; Yao et al., 2023; Liu et al., 2024a;b; Goel et al., 2024) have been proposed as methods for removing memorized information or otherwise nullifying the impact of a model's having been trained on problematic data. This could potentially introduce a pathway through which harms of reproducing inappropriate or copyright data could be addressed in cases where action was not taken during data curation or model training. Related methods for direct model editing (Mitchell et al., 2021; 2022a; Meng et al., 2022; Hernandez et al., 2023) to remove learned harmful concepts, through editing activations (Zou et al., 2023a; Turner et al., 2023), concept erasure (Ravfogel et al., 2022; Belrose et al., 2023), or targeted lesions (Li et al., 2023a; Wu et al., 2023), could provide alternative approaches to achieving these aims.

Open problems:

**Ensuring unlearning methods are robust and well-calibrated.** Machine unlearning involves an interplay between specificity of, and generalization from, concepts to be unlearned. In particular, methods that successfully generalize can aid in cases where the unlearning target is hard to precisely specify. However, generalization may open the door to unintended side-effects if it results in the removal of non-target concepts (Cohen et al., 2024b). A challenge to be addressed then is to ensure that methods for machine unlearning and model disgorgement are well-calibrated in that they successfully generalize to comprehensively remove target concepts, while avoiding the removal of benign concepts.

**Extending unlearning and model editing to cross-lingual and cross-modal models.** As trends towards multilingual (Üstün et al., 2024) and multi-modal (Yin et al., 2023) models continue, there will be a need to extend model unlearning and editing techniques to these models. Questions remain as to the efficacy of such techniques when applied in such cases (Si et al., 2023), for example, regarding whether models retain concepts in other languages, despite that concept having been unlearned in English.

**Evaluating the efficacy of unlearning and direct model editing techniques.** A further outstanding question is how the efficacy of unlearning attempts can be evaluated (Lynch et al., 2024; Shi et al., 2024). Evaluations should aim to assess not only whether the influence of the specified unlearning targets has indeed been removed and that model performance in other domains has not been adversely affected (Li et al., 2024b), but also identify potential *ripple effects* that may have resulted from an application of unlearning or model editing (Cohen et al., 2024b).

## 6.4 Deployment

---

**Example Research Questions**

---

87. How can the robustness of methods for detecting adversarial attacks be improved? (6.4.1)

88. What interventions are most effective for handling detected adversarial attacks at inference time? (6.4.1)

89. How can a model be made resistant to being fine-tuned for malicious tasks, while still allowing for benign fine-tuning? (6.4.2)

90. How can the request of dual-use system capabilities be reliably detected? (6.4.3)

91. How could authorization of user identity be used as a gate for dual-use model capabilities? (6.4.3)

---

### 6.4.1 Detection of Adversarial Attacks

Motivation: Adversarial attacks refer to deliberate attempts to exploit inherent model vulnerabilities in order to make the model behave incorrectly or harmfully, for example, by outputting offensive or toxic language (Lohn, 2020; Shayegani et al., 2023; Vassilev et al., 2024). In the case of language models, such attacks often take the form of modifying the user input so as to bypass any implemented safety filters. Some attacks are transferable across different models (Zou et al., 2023b) and defenses against adversarial attacks are typically narrow and brittle (Narayanan & Kapoor, 2024). The ability to detect such attacks could enable the application of targeted system-level defenses, such as halting or filtering system output, separate from relying solely on the underlying model's robustness to attacks. Furthermore, having empirical evidence on the frequency of attacks can help inform deployment corrections (Section 7.2) and threat models (Section 8.1).

While some system-level defenses against adversarial attacks exist, it is important to note that many such protective measures can only be implemented effectively in an application or deployment context (Narayanan & Kapoor, 2024). Though directly improving the adversarial robustness at the model level is a related active research area (Vassilev et al., 2024; Hu et al., 2024; Su et al., 2024) here we emphasize the closely related issue of being able to detect and handle potential adversarial attacks at inference time due to its relevance for governance interventions as mentioned above.

Open problems:

**Detecting adversarial inputs and outputs.** Being able to detect and classify user inputs to a model as potential adversarial attacks allows for filtering (Jain et al., 2023a; Aldahdooh et al., 2022) or preprocessing (Cohen et al., 2019; Nie et al., 2022; Kumar et al., 2023; Jain et al., 2023a; Zhou et al., 2024) of concerning inputs before being given to the model. Alternatively, model outputs could be filtered with the aim of detecting a model's response to adversarial attacks in order to remove them before reaching the user (Phute et al., 2024; Greenblatt et al., 2023). Current techniques for detection, however, can suffer from a lack of robustness themselves or may introduce significant latency for the user (Glukhov et al., 2024).

### 6.4.2 Modification-Resistant Models

Motivation: Post-deployment fine-tuning is a common method for user customization of language models, either through an API or locally in the case of downloadable models. Fine-tuning, and other post-training enhancements have been theorized to have an outsized impact on downstream performance (Davidson et al., 2023). However, just as fine-tuning can be used to customize a model for legitimate and beneficial use cases, it can just as easily be used to adapt a model for malicious purposes, often with small amounts of data (Jain et al., 2023b; Yang et al., 2023; Qi et al., 2023; Lermen et al., 2023; Zhan et al., 2024; Qi et al., 2024a;b). Fine-tuning attacks often aim to achieve similar aims to those of adversarial attacks above, though through customizing models directly, rather than by modifying input prompts. Having methods for preventing the customization of models for malicious use could reduce misuse risks associated with open-weight release,

thus expanding the range of potential deployment options and promoting the numerous benefits of more open release strategies.

Open problems:

**Preventing the modification of models for malicious tasks.** An open question is whether there exist technical methods that restrict a model's amenability to being fine-tuned (or modified through other methods) for harmful uses, while retaining the ability to be modified for benign uses (Rosati et al., 2024b; Peng et al., 2024). Potential methods may aim to raise the computational cost of fine-tuning on harmful data to prohibitive levels (Henderson et al., 2023b; Deng et al., 2024; Rosati et al., 2024a; Tamirisa et al., 2024) or make models resistant to learning from harmful data (Zhou et al., 2023b; Huang et al., 2024c; Hsu et al., 2024; Huang et al., 2024b). However, given the nascency of these techniques, future research could aim to establish their robustness in practice.

### 6.4.3 Detection and Authorization of Dual-Use Capability at Inference Time

Motivation: In the event that model assessments have flagged a system's competence in dual-use domains – that is, domains which can have both beneficial and harmful applications – model providers might need to avoid exposing these capabilities publicly by default in order to avoid misuse. However, completely removing these capabilities may not be feasible, or economically favorable due to the legitimate and beneficial use-cases, such as a cybersecurity professional using a system to aid in the identification and patching of software vulnerabilities. This differs from the discussion of adversarial attacks above, for which the aim of an attack is taken to be unambiguously harmful.

Open Problems:

**Detecting requests of dual-use capabilities.** Guarding against malicious uses of dual-use capabilities is currently imperfectly achieved by conducting safety fine-tuning so that the model refuses to respond to malicious requests. However, this approach is not robust to jailbreaks that pose as legitimate requests for such capabilities (Wei et al., 2023; Fang et al., 2024). An alternative approach could be to detect all requests for dual-use capabilities. This would allow for the application of separate methods for distinguishing between legitimate and malicious requests, such as independent classifiers trained to separate between legitimate and malicious intent, which may perform better than broad safety fine-tuning.

**Requiring authorization for dual-use capabilities.** Alternatively, authentication, for example as a certified cybersecurity expert, before accessing certain capabilities may be one way of managing the dual-use nature of general models. This may also have uses for allowing red-teamers or researchers to access such capabilities for research purposes (Longpre et al., 2024a). Such proposals are currently hypothetical, and so future work could aim to propose proof-of-concept demonstrations of how such an authorization scheme could be implemented in practice.

# 7 Operationalization

Previous sections have discussed concrete technical problems in relation to specific targets in the taxonomy. In contrast, the following two sections will discuss capacities that span across these targets, namely, *operationalization*, and *ecosystem monitoring*.

Operationalization entails the translation of ethical principles, legal requirements, and governance objectives into concrete strategies, procedures, and technical standards. It can also involve the harmonization of terminology and concepts across governance frameworks, for example, NIST's *Risk Management Framework Crosswalks* (NIST, 2023a). Without technical expertise, operationalization efforts may fail to capture the nuances and realities of AI systems, leading to ineffective or even counterproductive governance initiatives.

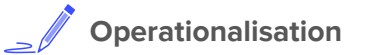

**Operationalisation**

- Translation of Governance Goals into Policies and Regulatory Requirements
- Deployment Corrections

Figure 6: Open problem areas in the *Operationalization* capacity

---

**Example Research Questions**

92. What system properties (if any) are the most reliable indicators of risk, and thus candidates for serving as regulatory targets? (7.1)

93. How can AI safety, reliability, and other technical requirements be standardized given an insufficient explanatory understanding of model behavior? (7.1)

94. What are general intervention and correction options if flaws with a model are identified post deployment? (7.2)

---

## 7.1 Translation of Governance Goals into Policies and Requirements

Motivation: Policies are often formulated with specific aims in mind, for example, to protect consumer safety, promote fairness, or ensure accountability. In cases of *rule-based regulation*, these aims must be translated into rules that "typically prescribe or prohibit a specific behavior" (Schuett et al., 2024). In many cases this act of translation will demand involvement of technical expertise to provide guidance on the feasibility of proposed rules, as well as the extent to which they achieve a policy's stated aims. For example, the goal of ensuring consumer safety may motivate the introduction of a licensing regime that mandates pre-market safety evaluations. Given the current robustness of such evaluations (see Section 3), this may not only fail to ensure safe and reliable products, but also create a false sense of security (Reuel et al., 2024b; Wu, 2023). For many governance efforts, concrete translations of goals into effective requirements and standards are still lacking (Guha et al., 2024; Pouget, 2023).

Open Problems:

**Identifying target dimensions for regulation.** Identifying technical dimensions that best align with governance priorities is an open challenge. For example, the current practice of using training compute[24] as a measure of risk may not always be suitable, given that smaller models can outperform larger ones with targeted training. Furthermore, modern AI systems are often the result of multiple data curation and training processes, and it is unclear whether compute expenditure from auxiliary processes should contribute towards

---

[24]Measured in *floating-point operations* (FLOP).

the final FLOP count. It's also unclear how measures of training compute should take into account techniques such as quantization and drop-out (Hooker, 2024).[25] Aside from training compute, are there other, more precise ways of defining which systems should be subject to regulation? How could such measures account for improved algorithms, and ways in which relevant capabilities can be increased after training (Davidson et al., 2023; Scharre, 2024)?

**Detailing and creating standards across the AI system life cycle.** While recent AI standard-setting efforts, such as the NIST Risk Management Framework (NIST, 2023b; 2024) and ISO guidance on AI risk management (International Organization for Standardization, 2023), provide valuable general principles, they often lack the technical specificity required for objective assessment of AI systems' compliance with safety and ethical requirements (Pouget, 2023). Additionally, though standard-setting bodies such as IETF, IEEE, ISO, and CEN-CENELEC, along with initiatives such as the Partnership on AI (Partnership on AI, 2023), are working to develop more detailed and verifiable guidance, many areas still require further technical expertise and research (Barrett et al., 2023). One such area is that of security-by-design for hardware, where further work is needed to implement standards across firms, including standards for multi-device attestation in a cluster and TEEs (Cybersecurity and Infrastructure Security Agency, 2023; Kelly et al., 2022, see also Section 6.2). Other goals, such as fairness, can be measured in various ways, and it remains unclear which metrics are most appropriate and effective in which contexts (Parraga et al., 2023; Caton & Haas, 2024; Chouldechova, 2017; Kleinberg, 2018). Finally, standardizing reporting for AI systems, such as the information included in model cards (Mitchell et al., 2019) or data sheets (Gebru et al., 2021), could increase the utility of such practices in governance contexts, motivating the question of what specific information should be included in standardized reports (Kolt et al., 2024; Bommasani et al., 2024).

## 7.2 Deployment Corrections

Motivation: In the event that flaws are identified in a deployed model, it would be beneficial to adequately respond to the identified risk. Such a scenario could occur either through the identification of previously unobserved capabilities in a deployed model, or through post-training enhancements such as fine-tuning (Davidson et al., 2023). O'Brien et al. (2023) refer to post-deployment responsive actions as "deployment corrections." While they explore this issue from an institutional perspective, for example providing recommendations on how this could be addressed through corporate governance structures and procedures, we see scope for much greater exploration from a technical perspective.

Open problems:

**Navigating the continuum of model corrections and interventions.** O'Brien et al. (2023) define five categories for deployment corrections: *user-based restrictions*, *access frequency limits*, *capability or feature restrictions*, *use case restrictions*, and *model shutdown*. Within each of these categories there are open questions regarding the feasibility of implementation. For example, *model shutdown* is a relatively extreme action to take upon discovery of a system flaw, and if carried out naïvely could risk major disruption to users, clients, and services that depend on the system in question. Thus, it would be beneficial to have methods in place for minimizing disruption to downstream services in the event that model shutdown is deemed necessary. Furthermore, model shutdown, along with deployment corrections that modify the underlying model, are at odds with the issue of providing model stability and backward-compatibility – features that are particularly relevant for ensuring the reproducibility and replicability of AI research (see Sections 4.3.1 and 5.3.2).

---

[25]Some recommendations are put forth in (Frontier Model Forum, 2024).

## 8 Ecosystem Monitoring

Due to the rapid pace of advancements in AI, coupled with uncertainty about future developments, AI governance needs to be forward-looking, future-proof, and adaptive (Guihot et al., 2017; Kolt, 2023; Reuel & Undheim, 2024). To fulfill this goal, decision-makers need to be aware of the multiplicity of stakeholders in the AI ecosystem and how they relate to each other, as well as general trends and potential impacts of current and future AI systems (Wansley, 2016; Ada Lovelace Institute, 2023; Whittlestone & Clark, 2021; Epoch, 2023). Collating and providing such information, which we refer to as *ecosystem monitoring*, can enable AI governance actors to make more informed decisions, better anticipate future challenges, and identify key leverage points for effective governance interventions.

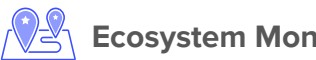 **Ecosystem Monitoring**

- Clarification of Associated Risks
- Prediction of Future Developments and Impacts
- Assessment of Environmental Impacts
- Supply Chain Mapping

Figure 7: Open problem areas in the *Ecosystem Monitoring* capacity

---

**Example Research Questions**

95. What risks, whether from intended or unintended harm, are associated with different (types of) systems? (8.1)

96. How do potential risks differ across domains? (8.1)

97. How can trends and/or properties observed in current systems be extrapolated to make predictions about future systems? (8.2)

98. How could developments in AI-specific hardware impact the governability of compute? (8.2)

99. What information about systems is needed to accurately assess the environmental impact of its development and deployment? (8.3)

100. Given the required information, how can the environmental impact of an AI system be accurately assessed? (8.3)

101. What technical methods can be implemented to create an auditable log of all actors and their contributions throughout the AI development process, from data collection to model deployment? (8.4)

---

### 8.1 Clarification of Associated Risks

Motivation: Understanding risks associated with the development and deployment of AI systems enables policymakers to prioritize governance efforts, allocate resources effectively, and determine the urgency of addressing specific risks (Whittlestone & Clark, 2021; Clark, 2023).

Open Problems:

**Developing better threat models for risks of AI.** While much prior work has intended to lay out taxonomies of risks and harms posed or exacerbated by AI systems (Critch & Russell, 2023; Hendrycks et al., 2023; Weidinger et al., 2022; Abercrombie et al., 2024; Hammond et al., forthcoming; Zeng et al., 2024;

OECD, 2023; Hoffmann & Frase, 2023; Turchin & Denkenberger, 2018; Grabb et al., 2024), detailed threat models have been relatively underexplored. One option for future research could aim to apply standardized risk management approaches, such as causal mapping (Eden et al., 2004; Ackermann et al., 2014), to gain greater clarity into if and how harms from AI may materialize, and where policy could intervene.

**Improving incident reporting and monitoring.** Additionally, developing improved systems for monitoring and reporting previous or ongoing incidents could not only allow for a more targeted response to ongoing harms, but also facilitate the identification of early warning signals for potential harms (Shane, 2024). AI incident databases have been developed by both the OECD and Partnership on AI, both of which log news articles detailing AI-related incidents (OECD.AI Policy Observatory, 2024; McGregor, 2020). Given that these databases rely solely on public sources, it is likely that only a subset of all incidents are included. In addition, they do not record all details about an incident such as model specifics or deployed guardrails, limiting the utility for analysis of what may have caused an incident. Open questions thus concern how non-public incidents can be reliably reported, as well as what technical information should be reported in order to facilitate meaningful analysis of incidents.

## 8.2 Prediction of Future Developments and Impacts

Motivation: Anticipating the trajectory and potential impact of AI systems may allow policymakers to proactively set governance priorities, determine the urgency of addressing specific issues, and allocate resources accordingly (Toner et al., 2023). Greater foresight would enable more adaptive and anticipatory approaches to AI governance, which are essential given the rapid pace of AI development (Reuel & Undheim, 2024).

Open Problems:

**Measuring and extrapolating from empirical trends.** Existing work has aimed to empirically measure trends in training compute (Sevilla et al., 2022) and algorithmic progress (Ho et al., 2024), among others (Epoch, 2023). Future work could aim to extend this effort by quantifying other trends that have not yet been addressed, such as usage patterns of AI in different industries, or assessing the accuracy of predictions based on the extrapolation of observed trends.

**Estimating a system's impact before deployment.** Estimating the impact of an AI system before deployment, including economic impacts (Eloundou et al., 2024), could help prioritize governance efforts. While such predictions could be aided by more developed threat models (see Section 8.1 above), research may also benefit from technical tools to safely and ethically experiment and simulate potential outcomes without causing harm.

## 8.3 Assessment of Environmental Impacts

Motivation: The environmental impact of AI systems extends across the entire AI life cycle (Metcalf et al., 2021; Luccioni et al., 2023b; Rakova & Dobbe, 2023), including both during training (Strubell et al., 2019; Patterson et al., 2022) and inference (Luccioni et al., 2023a). Having an accurate understanding of the end-to-end environmental impacts is crucial for policy initiatives, for example, to determine suitable incentives and penalties for encouraging AI developers to reduce the environmental costs associated with their systems.

Open Problems:

**Assessing the energy usage of training and hosting systems.** Open problems remain due to the logistical challenges of tracking energy consumption and carbon emissions across numerous dynamic system instances. Furthermore, current efforts struggle to take into account energy sources – a factor which can massively affect the overall impact assessment. Ongoing work aims to develop energy ratings for combinations of models and tasks, allowing users to make informed decisions about their system usage, taking into account the environmental impacts of their choice (Luccioni, 2024). Alternatively, tools such as *CodeCarbon*[26] provide developers with real-time estimates for the carbon emissions from running their code. Other work has focused on comparing compute cost on smartphones vs. the cloud (Patterson et al., 2024), best practices for training

---

[26]https://codecarbon.io/

models (Patterson et al., 2022) and comparing cost for different models (Luccioni et al., 2023a; Luccioni & Hernandez-Garcia, 2023).

**Assessing environmental costs of raw resources for building and running data centers.** Along with energy, environmental costs may come from other sources along the semiconductor and AI supply chains, for example, from mining and refining the rare earth minerals required for the manufacturing of semiconductors (Kuo et al., 2022; Ruberti, 2023). Additionally, large data centers used for training and hosting AI systems require great quantities of water as part of their cooling systems (Mytton, 2021). Future research could aim to provide in-depth end-to-end predictions of the environmental costs of constructing and maintaining data centers to inform policies aimed at reducing associated environmental impacts.

### 8.4 Supply Chain Mapping

Motivation: Mapping the AI supply chains can allow policymakers to better understand the complex ecosystem involved in the development and deployment of AI systems. By identifying key actors and processes at each stage of the supply chain, policymakers can target interventions at the most suitable point in the supply chain. Furthermore, existing export controls limiting chip exports to Russia and China have been marked by substantial enforcement difficulties (Allen et al., 2022), and analyses have suggested that AI chips are also likely to become targets for substantial smuggling operations (Grunewald & Aird, 2023; Fist & Grunewald, 2023). By understanding the flow of these resources, authorities can better combat the smuggling of chips and other hardware components.

Open Problems:

**Identifying supply chain components and actors.** Another area requiring further technical expertise is the identification and assessment of supply chain components. For example, in the context of liability and copyright law, tracking components and design choices made by different actors along the AI supply chain could enable courts to make more precise assessments of potential infringement responsibility (Lee et al., 2024; Longpre et al., 2024b). This granular understanding might be necessary as infringement can occur at multiple points: during data collection, model training, or output generation. If a model produces content resembling copyrighted material, determining liability may require tracing back through the supply chain to identify the source of infringement, whether in training data, model architecture, or generation prompt.

# 9 Conclusion

In this paper, we presented a broad overview of open problems in technical AI governance (TAIG) across six capacities and four governance targets. We provided a definition of TAIG, a corresponding taxonomy of the work that it entails, and an overview of sub-problems for each sub-area defined in our taxonomy. We also provided links to relevant literature and example research questions that technical researchers could tackle to help advance AI governance efforts.

We also showed that TAIG is not a monolithic problem – instead, it comprises interrelated sub-problems requiring distinct yet complementary approaches. Our taxonomy provides a structured framework for addressing such issues: Each open problem area presents unique technical bottlenecks, such as challenges in attributing model behaviors to training data, ensuring privacy-preserving access, and verifying system properties. By outlining these constraints, we highlight where governance aspirations are hindered by technical limitations and where technical research can drive progress in AI governance.

Our work further highlights that technical work does not exist in isolation. Instead, addressing many of these problems will require cross-disciplinary collaboration among technical researchers, policymakers, industry stakeholders, and civil society. No single discipline can fully resolve AI governance challenges, necessitating integrated expertise across computer science, law, political science, ethics, and economics. As AI systems become more capable and widely deployed, governance strategies must develop alongside technical advances, making TAIG an evolving field and ongoing priority.

**Acknowledgements**

LHa acknowledges the support of an EPSRC Doctoral Training Partnership studentship (Reference: 2218880). RS acknowledges support from Stanford Data Science, and an OpenAI Superalignment grant. NG acknowledges support from a Stanford Interdisciplinary Graduate Fellowship. YB acknowledges funding from CIFAR. AP acknowledges a the support of gift from Project Liberty. SK acknowledges support by NSF 2046795 and 2205329, NIFA award 2020-67021-32799, the Alfred P. Sloan Foundation, and Google Inc. DB, PB and AR acknowledge funding from Open Philanthropy and the Stanford Interdisciplinary Graduate Fellowship.

The authors would also like to acknowledge the early feedback received as part of a *Work-In-Progress* meeting, hosted by the Centre for the Governance of AI. We would particularly like to thank Jamie Bernardi, Ben Clifford, Augustin Godinot, John Halstead, Leonie Koessler, Patrick Levermore, Sam Manning, Matthew van der Merwe, Aidan Peppin, James Petrie, and Christopher Phenicie for detailed comments and insightful conversations. We further thank Dewey Murdick for his thoughtful and constructive feedback, which significantly improved the paper's rigor.

Finally, the authors would like to thank Beth Eakman and José Luis León Medina for support with copy-editing and typesetting, respectively.

# A   Appendix: Policy Brief

The increasing adoption of artificial intelligence (AI) has prompted governance actions from the public sector, academia, civil society, and industry. However, policymakers often have insufficient information for identifying the need for intervention and assessing the efficacy of different policy options. Furthermore, the technical tools necessary for successfully implementing policy proposals are often lacking. We introduce the field of **technical AI governance**, which seeks to address these challenges.

**Technical AI governance** refers to *technical analysis and tools for supporting the effective governance of AI*. We argue that technical AI governance can:

1. *Identify* **areas where policy intervention is needed** through mapping technical aspects of systems to risks and opportunities associated with their application;

2. *Inform* **policy decisions** by assessing the effectiveness and feasibility of different policy options; and

3. *Enhance* **policy options** by enabling mechanisms for enforcing, incentivizing, or complying with norms and requirements.

We taxonomize technical AI governance according to elements of the AI value chain: the inputs of **data**, **compute**, **models**, and **algorithms**, through to the **deployment** setting of the resulting systems. The figure below shows the key governance capacities that can be applied to each target.

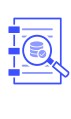

**Assessment**

Enables the understanding of system capabilities and risks, to allow for more targeted policy intervention.

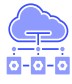

**Access**

Enables external research and assessment of AI systems, and the fair distribution of the benefits of AI.

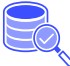

**Verification**

Establishes trust in AI systems and confirms compliance with regulatory requirements.

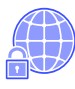

**Security**

Ensures the integrity, confidentiality, and availability of AI systems and guards against misuse.

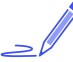

**Operationalization**

Bridges the gap between abstract principles and the implementation of norms and requirements.

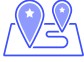

**Ecosystem Monitoring**

Enables anticipation of future challenges and identification of levers for governance intervention.

**Key Takeaways**

We highlight a number of the key takeaways within technical AI governance, including that:

- **Evaluations** of systems and their downstream impacts on users and society have been proposed in many governance regimes. However, current evaluations lack robustness, reliability, and validity, especially for foundation models.

- **Hardware mechanisms** could potentially enable actions including facilitating privacy-preserving access to datasets and models, verifying the use of computational resources, or attesting to the results of audits and evaluations. However, the use of such mechanisms for these purposes is largely unproven.

- The **development of infrastructure for enabling research into AI**, such as resources for conducting analyses of large training datasets or for providing privacy-preserving access to models for evaluation and auditing, could facilitate research that advances the scientific understanding of AI systems and external oversight into developers' activities.

- Research that aims to **monitor the AI ecosystem** by collecting and analyzing data on trends and advances in AI has already proven crucial for providing policymakers with the information needed to ensure that policy is forward-looking and future-proof.

We note that technical AI governance is merely one component of a comprehensive AI governance portfolio, and should be seen in service of sociotechnical and political solutions. A technosolutionist approach to AI governance and policy is unlikely to succeed.

**Recommendations**

Based on the above takeaways, we recommend:

1. **Allocating funding and resources** through open calls and funding bodies, to technical AI governance research, drawing on established expertise in adjacent fields;

2. That policymakers **collaborate closely with technical experts** to define feasible objectives and identify viable pathways to implementation;

3. That government bodies, such as AI Safety Institutes, **conduct in-house research on technical AI governance** topics, beyond their current focus on performing evaluations; and

4. That the future summits on AI, other fora such as the G7, the UN AI advisory body, and reports such as the *International Scientific Report on the Safety of Advanced AI*, **focus effort and attention towards technical AI governance**.

Please have a low bar for reaching out to Anka Reuel (`anka.reuel@stanford.edu`) and Ben Bucknall (`bucknall@robots.ox.ac.uk`) with any questions or comments.

# B   Appendix: Methodology

We grounded our methodology in established practices from governance and policy research. Taxonomies are widely used in these fields to organize complex domains into structured categories (Smith, 2002). A taxonomy is essentially a "formal system for classifying multifaceted, complex phenomena according to common conceptual domains and dimensions" (Bradley et al. (2007), based on Patton (2002)). Typologies and taxonomies can help bring clarity to ill-defined problem spaces (Sofaer, 1999; Smith, 2002). Following this previous literature and work, we adopted a taxonomy-based approach to systematically map the landscape of technical AI governance challenges.

As outlined in Section 1, previous AI governance research, in particular, has introduced such taxonomies and frameworks to categorize governance issues. For example, Dafoe (2018) divided the AI governance space into three clusters – technical landscape, AI politics, and ideal governance structures – to attempt comprehensive coverage of AI governance challenges. Likewise, Critch & Russell (2023) proposed a taxonomy of societal-scale AI risks organized by accountability (i.e., who the responsible actors are and whether harmful actions are deliberate or emergent). These prior efforts illustrate that there is no single 'correct' AI governance taxonomy; rather, their value lies in choosing organizing principles that reveal new insights or gaps (Bailey, 1994).

For our research, we began with a systematic search on Google Scholar and SSRN using the search terms *'Technical' AND ('Artificial Intelligence' OR 'Machine Learning' OR 'AI' OR 'ML') AND ('Governance' OR 'Regulation')* to identify academic literature targeting technical AI governance, which did not yield any search results based on paper titles or abstracts. We then expanded search efforts to include the terms *('Artificial Intelligence' OR 'Machine Learning' OR 'AI' OR 'ML') AND ('Governance' OR 'Regulation')* to identify goals, functions, and challenges of AI governance more broadly, not just focused on technical work. In addition, we built on efforts by Reuel et al. (2024b) and compared existing regulatory and governance aims and the technical state of the art to identify gaps and governance challenges that could be addressed with technical advancements (these insights were later also integrated into the 'Motivation' sections for each open problem area in Sections 3 to 8).

Following this initial literature review, we generated a list of candidate capacities and open problems. We red-teamed this preliminary taxonomy with a focus group of five experts across AI and governance. Based on their feedback, which also referenced the work by Bommasani et al. (2023b), we added the *target* dimension to the taxonomy. We applied this mixture between deductive and inductive approaches throughout the whole taxonomy development process, in line with best practices for taxonomy development (Bailey, 1994; Nickerson et al., 2013). Our taxonomy then went through multiple rounds of expert- and literature-driven refinements. We iteratively discussed and adjusted capacity definitions in a series of interviews and focus groups with the authors, who are all researchers or practitioners in (AI) governance or relevant technical fields, and external experts, involving over 30 researchers and practitioners across all capacities, targets, and identified open problems areas. In these sessions, we critically examined whether each proposed category was conceptually distinct and collectively exhaustive of the domain's key issues and whether we missed any open problems. The resulting feedback led to taxonomy adjustments (e.g., clarifying the scope and definition of the *Operationalization* capacity) until we converged on a final taxonomy structure. Whenever a new capacity was added (including for the initial set of defined capacities), we also conducted a literature search on Google Scholar and SSRN with the terms *('Artificial Intelligence' OR 'AI') AND '[Capacity]'* to validate it, identify existing research in this area, and to expand on open problems in the identified area. We then again requested expert feedback on gaps, previous literature, open research questions, and missing open problems for the newly-added capacity.

We validated our final taxonomy's relevance by comparing it with previous classifications of issues in AI governance research and AI risk taxonomies, looking for overlaps and differences. We compared our open-problems categories to challenges raised in Dafoe (2018); Lobel (2024); Shelby et al. (2023); Raji et al. (2022a); Google (n.d.); Critch & Russell (2023); Weidinger et al. (2022); Birkstedt et al. (2023), confirming that all major governance challenge areas identified by these authors that could have a technical component have a place within our two-dimensional framework. In fact, many open problems we identify (such as evaluating

model risks or securing compute infrastructure) are technical prerequisites to addressing the higher-level governance challenges described in these prior works. For example, the challenge of governance failure due to inadequate oversight (Uuk et al., 2024) can be partially addressed by our verification capacity, and the risk of AI misuse (Critch & Russell, 2023) could be mitigated by advances in security and assessment, two of our capacities which are also governance mechanisms highlighted by McCormack & Bendechache (2024) and Lobel (2024). This cross-validation across capacities gave us confidence that our taxonomy is neither missing well-known challenge domains nor duplicating existing taxonomies without added value. Instead, it synthesizes and builds on prior efforts: for instance, where Critch & Russell (2023) focuses on *who* is responsible for risks, and Shelby et al. (2023) catalog harms by *impact type*, our taxonomy focuses on *how* technical work can proactively support governance.

Finally, we acknowledge our taxonomy's limitations. It is expert-defined and thus subject to unique perspectives and experiences. Furthermore, validation is challenging, given that there is no ground truth for our taxonomy to test against. It is further constrained to technical challenge areas by design, meaning it leaves out many important socio-technical or institutional governance issues (e.g. labor impacts, international dynamics) highlighted by others (Dafoe, 2018; Lobel, 2024; Shelby et al., 2023; Raji et al., 2022a; Critch & Russell, 2023; Weidinger et al., 2022).

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
