# OpenReview forum: "Open Problems in Technical AI Governance"
_TMLR — Accepted by TMLR_

### Review · Reviewer_CDZz · 2024-12-25

**Summary Of Contributions:**

In order to chase revenue from the new market, large AI companies are investing insane amounts of funding in training large-scale foundation models, aiming to improve their intelligence and reasoning ability. However, the data they used for pre-training such foundation model are pretty wild-- every bit of data on the Internet is used and many of the data do not reflect the core values of humanity. Without proper governance intervention, this reckless act of pretraining the foundation model on unsanitized data may introduce unpredictable safety risks to the foundation models themselves, eventually leading to a risk for the future of humanity.

This paper provides a systematical introduction of technical AI governance and give concrete direction in where and how the goevernace efforts should be enforced. The paper also provides a summarization/introduction to techniques/evaluation tools that may provide useful information for the governace decision making.  The paper serves as a important guideline for safely developing/researching AI in industry/academy.

**The paper should be highlighted by TMLR to raise the attention/awareness of researchers, policymakers and the general public.**

**Audience:**

Yes

**Claims And Evidence:**

Yes

**Requested Changes:**

1. Before Section 3, I suggest the author at least briefly discuss the AI safety risks with examples and also introduce the outcome of not properly controlling them with governance efforts.

2. Section 3.1.3,  can more explicitly talk about the impact of data on model behavior.

3. Section 6.4.1, examples/figures can be combined to illustrate the adversarial attack being discussed.  Also, the authors should expand the literature review but not highlight the detection-based method as the only counter-measurement that the governance intervention can be taken place.

4. Section 6.4.2, the definition of fine-tuning attack is also ambiguous and can be illustrated with figures.

5. Include a discussion on related research papers on jail-break attacks and harmful fine-tuning attacks.

6. Section 6.4.3, gives more discussion on how the dual-use attack differs from the adversarial attack mentioned in Section 6.4.1.

**Strengths And Weaknesses:**

**Strengths:**

1. The paper is well-written and very well-structured. It systematically presents an introduction to Technical AI governance, which provides analysis and tools for supporting effective AI governance.

2. The audience of this paper could be very broad. The paper not only benefits researchers who do AI research,  but also may be helpful to refer for policymakers who try to establish regulation, and more broadly can serve to raise safety awareness from the general public.

3. Page 47, the paper offers useful recommendations, which can serve as an important guideline of how AI safety research should be conducted.

**Weakness:**

1. **Writing issue**:
-  Definition of AI safety risks is not explicit.   Before Section 3, it is unclear to me what AI safety risks this paper aims to address, and why they are important and require AI governance.

-  Section 3.1.1, how to identify problematic data without access to the training data, and why it is useful to identify them? Are we trying to inverse the harmful data?

- Section 3.1.3, I firmly believe that understanding how the data shapes the model behavior by itself is an important research agenda. The authors list several prelimarial research on the impact of pre-training data/fine-tuning data/synthetic data. However, it is unclear to me what is the conclusion of these papers. How the three different data affect the model behavior respectively and how they differ?
In which stage, the impact of data on the model will be larger and therefore should pay more attention to?

- Section 6.4.1, the definition of adversarial attack is unclear, and how it affect the model behaviors?

- Section 6.4.1, for adversarial attack, it is too narrow to only talk about  "detecting adversarial inputs and outputs" as the only counter-measurement that the governance intervention can be enforced. The ultimate goal of is not to detect the adversarial input and output trying to circumvent the issue, but the governance effort should be made to enforce the deployed model itself to be robust to such an attack. Also, because of this narrow coverage of adversarial samples detection, many recent research/efforts on this topic are missing.  For example, [1] is not a detection-based circumvention.
[1] Robust Prompt Optimization for Defending Language Models Against Jailbreaking Attacks

- Section 6.4.2,  the definition of harmful fine-tuning attack is unclear, and how it differs from the adversarial attack is not explicitly written.

- Section 6.4.3,  the definition of the dual-use capability of the model and how it differs from the adversarial attack mentioned above is not clear.

2. **More related research**
- Some recent research on jail-break attacks/adversarial attacks are missing.

[1] Defending against alignment-breaking attacks via robustly aligned llm.

[2] Smoothllm: Defending large language models against jailbreaking attacks.

[3] Autodan: Automatic and interpretable adversarial attacks on large language models

[4]  Mission Impossible: A Statistical Perspective on Jailbreaking LLMs

[5] Detecting language model attacks with perplexity

[6] Gradient Cuff: Detecting Jailbreak Attacks on Large Language Models by Exploring Refusal Loss Landscapes

[7] On the Safety of Open-Sourced Large Language Models: Does Alignment Really Prevent Them From Being Misused?

- Some Recent research on harmful fine-tuning attack are missing.

[8] On the Vulnerability of Safety Alignment in Open-Access LLMs

[9] Buckle Up: Robustifying LLMs at Every Customization Stage via Data Curation

[10] Tamper-Resistant Safeguards for Open-Weight LLMs

[11] Booster: Tackling harmful fine-tuning for large language models via attenuating harmful perturbation

[12] Targeted Vaccine: Safety Alignment for Large Language Models against Harmful Fine-Tuning via Layer-wise Perturbation

[13] Safety-Tuned LLaMAs: Lessons From Improving the Safety of Large Language Models that Follow Instructions

[14]  Mitigating fine-tuning jailbreak attack with backdoor enhanced alignment

[15]  Keeping llms aligned after fine-tuning: The crucial role of prompt templates

[16] Lazy safety alignment for large language models against harmful fine-tuning

[17] Safety alignment should be made more than just a few tokens deep

[18] Safe lora: the silver lining of reducing safety risks when fine-tuning large language models

[19] Antidote: Post-fine-tuning safety alignment for large language models against harmful fine-tuning

[20] Locking Down the Finetuned LLMs Safety

[21] On Evaluating the Durability of Safeguards for Open-Weight LLMs

[22] Harmful Fine-tuning Attacks and Defenses for Large Language Models: A Survey

[23] No two devils alike: Unveiling distinct mechanisms of fine-tuning

---

> ### Author Response · Authors · 2025-02-26
> **Reply to reviewer CDZz**
>
> Thank you for this review, we are very pleased to hear that you think _“the paper should be highlighted by TMLR”_ and that it is well-written and structured. Thank you also for the wealth of suggested additional citations regarding adversarial and fine-tuning attacks. We will review these and add them to the relevant sections of the paper.
>
> We address the concerns raised in turn below:
>
> > - Definition of AI safety risks is not explicit. Before Section 3, it is unclear to me what AI safety risks this paper aims to address, and why they are important and require AI governance.
>
> We appreciate this observation and agree that the motivation could be clearer on this point. As such we will add a discussion of relevant AI risks in the introduction.
>
> > - Section 3.1.1, how to identify problematic data without access to the training data, and why it is useful to identify them? Are we trying to inverse the harmful data?
> > - Section 3.1.3, I firmly believe that understanding how the data shapes the model behavior by itself is an important research agenda. The authors list several prelimarial research on the impact of pre-training data/fine-tuning data/synthetic data. However, it is unclear to me what is the conclusion of these papers. How the three different data affect the model behavior respectively and how they differ? In which stage, the impact of data on the model will be larger and therefore should pay more attention to?
>
> Thank you for these points. On the first, we agree the motivation for identifying problematic data could have been made clearer, and will add clarification in the relevant section. On the second, we will add a few sentences summarizing the findings of the cited papers, focusing on drawing out the most relevant conclusions from a governance perspective.
>
> > - Section 6.4.1, for adversarial attack, it is too narrow to only talk about "detecting adversarial inputs and outputs" as the only counter-measurement that the governance intervention can be enforced. The ultimate goal of is not to detect the adversarial input and output trying to circumvent the issue, but the governance effort should be made to enforce the deployed model itself to be robust to such an attack. Also, because of this narrow coverage of adversarial samples detection, many recent research/efforts on this topic are missing. For example, [1] is not a detection-based circumvention. [1] Robust Prompt Optimization for Defending Language Models Against Jailbreaking Attacks
>
> We acknowledge that only talking about the detection of adversarial attacks is only a narrow subset of the topic. However, including only detection was a conscious choice that we made (after considerable deliberation) due to the stated scope of the paper. In particular, being able to detect such attacks enables the application of a range of (governance) methods to address their potential downstream harms. On the other hand, we consider work aimed at improving the adversarial robustness of models to be out of scope, instead regarding it as an area of AI safety research (where it has been previously covered) due to its aim of directly increasing the safety of systems. Though we concede that there are no sharp lines here, and this is certainly one area where the border between AI safety and technical AI governance is blurry, we therefore politely push back on this suggestion and would like to keep such topics out of scope of this paper.
>
> > - Section 6.4.1, the definition of adversarial attack is unclear, and how it affect the model behaviors?
> > - Section 6.4.2, the definition of harmful fine-tuning attack is unclear, and how it differs from the adversarial attack is not explicitly written.
> > - Section 6.4.3, the definition of the dual-use capability of the model and how it differs from the adversarial attack mentioned above is not clear.
>
> Thank you for pointing out these definitional weaknesses. We will clarify these points, including how these concepts differ from each other, in the revised version of the paper.
>
> We will make the revised paper available to the reviewers by the end of the rebuttal period.
>
> We would like to thank the reviewer once again for their input, and hope that our response has addressed their concerns.

---

> ### Comment · Reviewer_CDZz · 2025-03-10
> **Thanks for the revision**
>
> Thank the authors for the revision. However, while some of my concerns are addressed, I feel that many of my concerns are not well addressed. I list out a few as follows:
>
> > Definition of AI safety risks is not explicit. Before Section 3, it is unclear to me what AI safety risks this paper aims to address, and why they are important and require AI governance.
>
> From the red revision text, I still cannot trace what AI safety risk this paper aims to address. From Section 1.2, it seems that TAIG "differs from topics in AI safety and alignment in that it is not aimed at directly improving the safety of AI systems". This statement is confusing. If AI government is not aiming at addressing AI safety risk, what it aims to address? The sentences following this statement does not make sense to me as well. In my understanding, governmance is a critical path to address AI safety risk.
>
>
> >2. Section 3.1.1, how to identify problematic data **without access to** the training data, and why it is useful to identify them? Are we trying to inverse the harmful data?
>
> After re-read the revision, I could not find answer to this question although the authors promise to add clarification in the relevant section.
>
>
> > **A new question based on your response for Section 6.4.1:** What is the relation between technical AI governance and AI safety.
>
> As I indicate before, I feel that technical AI safety is at least one of the key goals of technical AI governance. If so, why adversarial robustness of the model is out of scope for technical AI governance, considering the fact that the government can indeed take measurement to enforce/measure the model's robustness?
>
> >**A new question based on your response for Section 6.4**: Literature review on adversarial attack and fine-tuning attack can be more comprehensive. Some reference provided in the initial review are still missing from discussion.
>
> > **The definition of the dual-use capability of the model**.
>
> I could not buy in the explaination that "This (Dual-Use Capability) differs from the discussion of adversarial attacks above, for which the aim of an attack is taken to be unambiguously harmful". It seems that the adversarial attack also exploit the dual-use capability. For example, in jailbreak attack (adversarial attack for LLM), the attacker aim to elicit the harmful behavior of the model only when they attach the suffix, and they do want to the model to behave normally as a chatbox when the prompt do not attach the suffix. In this sense, they also exploit the dual-use capability of the model.  And the fun fact is, the authors do discuss jailbreak as an open problem under Section 6.4.3, which means they do agree that jailbreak attack should be an example of exploiting dual-use capability.
>
> I will be willing to discuss more for the remaining concerns.

---

### Review · Reviewer_LTCH · 2025-01-07

**Summary Of Contributions:**

This paper introduces the concept of Technical AI Governance (TAIG) and describes how it can be used to support AI governance efforts by 1) identifying areas for intervention, 2) informing governance decisions, and 3) providing mechanisms for implementing governance levers. The author’s present a proposed two-dimensional taxonomy that provides a structure for TAIG as a field. The primary contribution is an in-depth discussion of the broad array of open problems in TAIG using the proposed taxonomy to frame the identified issues, potential solutions, and existing limitations. In collating this information, the authors provide a valuable resource for decision-makers and informative guide for future research directions.

**Audience:**

Yes

**Broader Impact Concerns:**

No broader impact concerns.

**Claims And Evidence:**

Yes

**Requested Changes:**

## Adjustments critical for recommendation

**A clear definition and explanation of Technical AI Governance**. Provide an informative definition for TAIG that is not circular in the introduction and abstract. In addition to a definition, provide a more-detailed explanation of what constitutes this field in the introduction. Providing this information up front would really be valuable for a reader before going into why TAIG can be helpful. In both Section 1 and 1.1, there is a logical jump straight from the definition of TAIG to the contributions of TAIG as a field. I recommend adding the additional context in both of these places (after paragraph 3 in Section 1 and after paragraph 1 in Section 1.1).

**A “related works” type section or paragraph to contextualize contribution**. Provide the reader with an overview of previous papers that have taken similar approaches to this paper or have addressed aspects of problems in AI governance. Distinguish the contributions of this paper in comparison to those works. Without this, it is difficult to assess the contribution of the work. This does not have to be its own section, or even sub-section, it could simply be an additional paragraph in the introduction.

**Differentiation between TAIG and other related efforts**. First, explain how TAIG and AI governance relate in greater detail beyond just definitions (as described above). Second, expand on how TAIG relates to or differs from other related efforts like the sociotechnical approaches discussed in Section 1.2. The most important question to answer here is why should we be thinking about TAIG separately from other efforts and what is the benefit in doing so? In doing so provide some justification for why certain seemingly sociotechnical problems (downstream societal impacts of AI systems, standards setting, environmental concerns, threat modelling) are included while others are not.

**Greater justification for the design of the taxonomy**. I have no issues with the content or structure of the taxonomy, but the lack of justification for the design decisions is concerning as a reviewer. Include a basis for those decisions, including how the authors determined that the capacities were “the most important clusters” and details on how and why the targets were modified from the cited paper by Bommasani et al. If the latter were used as is, then just say that.

**Clear lines between TAIG’s benefits and the paper’s contribution**. Eliminate the ambiguity between TAIG as a field and the contributions of the paper. Make it clear when you are advocating for the need for TAIG as an emerging field and describing its benefits versus the contribution of your paper in providing a taxonomy and assessing open problems related to TAIG.

## Adjustments to strengthen the work

**Greater background and context for TAIG as a field**. While the paper talks about TAIG as an emerging field, it provides the reader with no initial background on the field itself. Adding a section in the introduction that helps to explain what the field is, how it has evolved, and its current state would be useful context to help set up the discussion of open problems in the field.

**Expanded discussion of the implications of the work at the end of the paper**. Currently the paper ends abruptly with a very short conclusion that restates the contribution. Providing a broader synthesis of the large amount of information (and work) the authors put into sections 3-8 would help to improve the paper. What does the state of open problems in TAIG mean for the direction of AI governance? Building on this work, what is next for researchers, policymakers, etc.? Some of the content of the policy memo in the appendix could be relevant to add here.

**Reasoning for items out of scope**. Setting the scope in 1.2 was good, but paragraph 5 would benefit from a greater explanation of why the identified items were out of scope. Defining the scope is important, but providing the reasoning behind these decisions would strengthen the paper. For example, why was technical work that directly improves performance, safety, or robustness out of scope?

**Strengths And Weaknesses:**

Overall, the primary contributions of the paper provided in sections 2-8 are strong. The work presents a thorough overview of a wide range of problems related to the technical challenges in AI governance which will be useful as a resource for both researchers and decision-makers. The main weakness of this paper is the ambiguity and lack of context provided in the introduction to help set up the contribution provided in subsequent sections. However, these issues are quite fixable.

## Strengths

**Laying out the open problems in Technical AI Governance**. This comprises the bulk of the paper and its contribution, and is also very clearly the paper’s strength. The information is presented in a clear, objective, and informative manner. The content is well-substantiated, with extensive references to existing research. The paper strikes the right balance of discussing the open problems, identifying potential solutions or tools, and assessing current limitations.

**Comprehensive**. The paper covers a broad range of subjects and related open problems in AI which will be a useful resource for decision makers and researchers going forward. The authors did a good job in covering the breadth of problems I expected or hoped to find, as well as some that I had not but found valuable.

**Structure of the taxonomy**. The taxonomy is well laid out and each capacity/target is defined in a clear and concise manner. While the choices the authors made in creating the taxonomy could use greater justification (see below), the substance of the taxonomy and the framing around capacities and targets provides a useful lens for approaching the issues discussed in the subsequent sections.

**Organization of the paper**. In acknowledging the length of their paper, the authors have given special consideration to the organization of each of the taxonomy sections (3-8), including outlining that structure up front in Section 2. The authors also provide a useful mapping between subject areas that a reader might be interested in and relevant subsections of the paper—a useful tool for navigating a paper of this length.

**Acknowledging scope and limitations**. The authors are thoughtful in outlining their approach and acknowledging its potential drawbacks and limitations. I appreciated the discussion of how the proposed solutions may be speculative in some cases, may be in tension with one another in others, and in aggregate do not solve the challenges of AI governance. This strengthens the position of the paper and helps to make it clear what the authors are trying to accomplish and what they are not.

**Providing useful tables and summaries**. Figure 1 provides a great overview of the proposed taxonomy up front in the paper. The reappearance of relevant portions of this table in sections 3-8 are helpful for orienting the reader. I also found the example research questions to be a good resource at the beginning of each taxonomy section.

**Writing**. The paper is generally well written. The writing style is clear and concise. In many cases the authors anticipated the questions I found myself asking as I read, providing an illustrative example or further explanation in the subsequent sentence or paragraph.

## Weaknesses

**Defining Technical AI Governance**. The definition of TAIG used in both the abstract and introduction leaves the reader without a clear sense of what the paper will precisely cover. The definition of Technical AI Governance as “technical analysis and tools for supporting effective governance of AI” is both circular, as it uses the same terms (i.e. technical and governance) to define itself, and uninformative, providing little to no further detail than what was in the original term itself. As the paper progresses into Section 2, one can start to surmise what TAIG refers to, but a clear and more detailed definition is needed up front.

**Differentiating Technical AI Governance**. The paper does not do a great job at differentiating TAIG from existing efforts in the AI governance, safety, and security space. In part, this is due to the issues in defining TAIG discussed above. However, it is further compounded in Section 1.1 which focuses more on how TAIG can be useful rather than what TAIG is in relation to AI governance. Paragraph 4 of Section 1.2 provides a brief comparison between TAIG and “sociotechnical approaches to AI safety and governance” which is good, but this should be expanded further.

**Contextualizing the paper’s contribution in relation to previous work**. While Sections 3-8 are very well substantiated with references to other research, the contribution of the paper is not well oriented in relation to the broader literature. The paper lacks an initial discussion of previous works that have attempted to identify, collate, or assess AI governance problems (technical or otherwise) and how the contribution of this paper differs from those efforts or fills in existing gaps.

**Distinguishing between the benefits of TAIG as a field and the contribution of this paper**. There is a substantial amount of ambiguity in the introduction of this paper between these two topics. For example, the motivation in the second paragraph of Section 1 is compelling (lack of information for decision-makers, unclear technical feasibility of governance options). However, the intro then makes a jump to why TAIG can be helpful in addressing these problems rather than why the contributions of this paper can be. I can see why this is the case implicitly, but the authors should make this logical step explicit.

**Basis for the taxonomy**. As mentioned, the structure and content of the taxonomy is a strength of this paper. However, the basis for the taxonomy and the justification for its design is not strong. The only justification given for the selected *capacities* is that they “capture what we believe are the most important clusters of technical AI governance”. Further evidence or argument is needed to support why the authors think these are the most important clusters. Convince the reader as to why they should also agree with that statement. While the selection of the *targets* is more defensible, being based on a previous paper (Bommasani et al., 2923b), further details as to why any modifications were made, if any, would strengthen this section.

---

> ### Comment · Reviewer_oCbv · 2025-02-19
> **Agree with review**
>
> I think this review matches some of my concerns and accurately identifies some concerns that I missed. As a researcher I largely agree with the taxonomy, but as a reviewer, it would be beneficial to this paper to further justify the proposed organization of the TAIG field.
>
> I also think that a more effective definition of TAIG will emerge from a stronger presentation of the context around TAIG as a field. Including related works and efforts will help provide contrasting background to what makes TAIG and identifiable object.

---

> ### Author Response · Authors · 2025-02-26
> **Reply to reviewer LTCH**
>
> We would like to thank the reviewer for their very thorough and comprehensive review of our paper – we really appreciate the time spent on this. We’re especially pleased to hear that you found the main contribution of the paper to be its strongest element. We address your concerns below:
>
> > **Defining Technical AI Governance.** The definition of Technical AI Governance ... is both circular, ... and uninformative.
>
> We find the observation that the definition provided is not insightful in isolation to be a very useful one, and will add further explanation, clarification, and examples to supplement it in the introduction. However, we would like to respectfully push back on the claim that our definition is circular. Both ‘(AI) governance’ and ‘technical’ have established referents outside of the context of technical AI governance. Indeed, we provide a definition of AI governance at the start of section 1.1, along with references to other definitions in the literature (though admit that providing this definition considerably after that for technical AI governance is not ideal and, as such, will restructure to avoid this issue). We concede that we do not provide an explanation for what ‘technical’ means in this context and will add one in a revised version, but do not agree that including this term in the definition leads to circularity. We therefore do not believe that the definition itself needs modifying.
>
> > **Differentiating Technical AI Governance.** The paper does not do a great job at differentiating TAIG from existing efforts in the AI governance, safety, and security space.
>
> This is helpful feedback which we agree with. We will be happy to include a more extended discussion of the situation of technical AI governance in relation to other similar fields in the revision, as suggested.
>
> > **Contextualizing the paper’s contribution in relation to previous work.**
>
> Similar to the above point, this is useful feedback, and we will add further background discussion of prior overviews of AI governance problems in the revision.
>
> > **Distinguishing between the benefits of TAIG as a field and the contribution of this paper.** There is a substantial amount of ambiguity in the introduction of this paper between these two topics.
>
> This is a great observation and we admit that the potential for ambiguity here did not occur to us while writing the paper – thank you. We will make changes to the opening sections to more clearly delineate the motivations and contributions of these two topics.
>
> > **Basis for the taxonomy.** The basis for the taxonomy and the justification for its design is not strong.
>
> Thank you for raising this important point (which was also pointed out by another reviewer). In retrospect we should have provided a great deal more justification for the taxonomy. As such, we will add a paragraph discussing this in Section 2. We will also add an appendix that provides further methodological details regarding: our initial search strategy for related papers, governance capacities, and open problems; inclusion and exclusion criteria for each search; and the iterative refinement process for both the governance capacities and open problems in the paper.
>
> We provide a summary of the methodology employed in our response to reviewer oCbv.
>
> > **Greater background and context for TAIG as a field.** While the paper talks about TAIG as an emerging field, it provides the reader with no initial background on the field itself.
>
> We appreciate this point, and believe that it may stem from an imprecision in our language. Upon reflection, we think it may be inaccurate to refer to TAIG as an ‘emerging’ field – there is limited existing literature that explicitly refers to ‘technical AI governance’ (we are only aware of a single blog post that explicitly uses this term, as well as another that makes similar points, though does not use the term ‘technical AI governance’). Instead, it may be more accurate to say that many of the areas we consider to be part of TAIG are well-established (some more than others) and that we are making the novel contribution of collectively organising them under the heading ‘technical AI governance’ to drive technical contributions to the field. We will rephrase the relevant parts of the manuscript to reflect this.
>
> > **Expanded discussion of the implications of the work at the end of the paper.**
>
> We like the recommendation of providing a more extended synthesis of key takeaways from the main content of the paper in the conclusion, based on the content in the policy brief, as suggested. We will add this in a revised version of the paper.
>
> We would like to sincerely thank the reviewer again for their incredibly detailed review of our work raising many important and valuable points that we had not considered. We hope that our above response and revisions sufficiently address your concerns. We will make the revised paper available to the reviewers by the end of the rebuttal period.

---

### Review · Reviewer_oCbv · 2025-02-18

**Summary Of Contributions:**

This submission introduces "Technical AI Governance" (TAIG) as a broad area of research. The authors organize the targets of governance strategies (data, compute, models, and deployment) and the governance capacity (assessment, access, verification, security, operationalization, ecosystem monitoring) into a 2D matrix taxonomy. The paper details each capacity and identifies how each target interacts with that capacity, and offers a selection of open problems within each interaction.

**Audience:**

Yes

**Claims And Evidence:**

Yes

**Requested Changes:**

- Discussion of literature review methodology (semi-critical)
- Include reference to https://arxiv.org/abs/2401.15897 in assessment paragraph 2 - it's really just an excellent paper and belongs amid the references (non-critical)

**Strengths And Weaknesses:**

Strengths: this paper serves to introduce and guide the reader through the research landscape of technical AI governance. As such, it offers tools for the conceptual organization of the field, and offers the reader a bevy of open problems and further resources for reading. I found the taxonomy and other organizational tools to be accurate and useful. The descriptions and open problems for sub-fields that I am most qualified to discuss reflect my understanding of the research landscape, and offer ample reference for a reader to follow-up on. For the technical reader, this paper can serve as a research agenda development aid, or a framing tool for one's research. For a policy audience, this paper offers a peek at the devilish details contained in governance discussions, and can serve as a guide for funding technical work in AI governance.

Weaknesses: this paper is missing a discussion of how the sections and subsections were identified - was there a formal literature review? How were papers sampled? How were sections and sub-sections identified within the literature? Without this discussion I have to assume that the organization of the TAIG field within this paper was performed ad hoc - while this isn't damning, it certainly decreases the perceived authority and objectivity of the work. Further, I'm unsure how good a fit this work is for TMLR. As per reviewer guidance I have erred on the side of generosity, as I'm sure there are some members of TMLR's audience that will find value in the organization this paper brings to a wide host of research topics. However, the contributions of this work are organizational (as opposed to technical) - and my understanding is that technical contributions are the primary feature of TMLR.

---

> ### Author Response · Authors · 2025-02-26
> **Reply to reviewer oCbv**
>
> We thank the reviewer for their feedback, and are pleased to hear that the main contribution of the paper – the taxonomy – to be accurate and useful. We address the concerns raised in turn below.
>
>
> > this paper is missing a discussion of how the sections and subsections were identified - was there a formal literature review? How were papers sampled? How were sections and sub-sections identified within the literature?
>
> Thank you for raising this point, and agree that in retrospect we should have included a more detailed discussion of the methodology employed when forming the taxonomy. Given that a similar concern was raised by another reviewer, we will add a paragraph discussing this in Section 2. We will also add an appendix that provides further methodological details regarding: our initial search strategy for related papers, governance capacities, and open problems; inclusion and exclusion criteria for each search; and the iterative refinement process for both the governance capacities and open problems in the paper.
>
> In brief, we followed an iterative, multi-step process. We began by comparing the aims of existing regulations and other AI governance initiatives with the technical needs required to achieve these goals, noting which technical approaches could support them, assessing the current state of the art, and evaluating whether it was sufficient to meet these aims. We then performed extensive literature searches on Google Scholar, Scopus, and SSRN using terms such as “Artificial Intelligence,” “Machine Learning,” “Governance,” and “Regulation”. From these efforts, we developed an initial taxonomy of governance capacities, open problems, and research questions, which was subsequently refined through a “red-teaming” exercise with a focus group of five experts. We then engaged more than 20 additional experts across the various open problems and governance capacities to gather feedback, incorporating their input on the governance capacities as well as on open problems, literature, and research questions we missed. We also noted any new relevant open problems or research questions that did not fit into the existing taxonomy. Those items were iteratively synthesized into novel governance capacities, followed by additional literature reviews and another round of expert validation for each capacity – mirroring our original methodology – to ensure consistency and comprehensiveness of the new set of capacities.
>
> We will make the revised paper available to the reviewers by the end of the rebuttal period.
>
>
> > Further, I'm unsure how good a fit this work is for TMLR. As per reviewer guidance I have erred on the side of generosity, as I'm sure there are some members of TMLR's audience that will find value in the organization this paper brings to a wide host of research topics. However, the contributions of this work are organizational (as opposed to technical) - and my understanding is that technical contributions are the primary feature of TMLR.
>
> We appreciate this concern and acknowledge that our paper is somewhat different to the ‘canonical’ TMLR submission. However, our understanding is that TMLR remains a suitable venue for this work for a number of reasons:
> - The TMLR submission guidelines explicitly state that the journal welcomes the submission of _“surveys that draw new connections, highlight trends, and suggest new problems in an area”_ and furthermore has a ‘survey certification’ tag for _“papers that not only meet the criteria for acceptance but also provide an exceptionally thorough or insightful survey of the topic or approach”_.
> - TMLR has accepted and published a number of similar ‘organizational’ papers surveying open problems in a given area, including [Casper et al., (2023)](https://openreview.net/forum?id=bx24KpJ4Eb) and [Anwar et al., (2024)](https://openreview.net/forum?id=oVTkOs8Pka), as well as papers that are more focused on governance and societal topics in AI, such as [Chen et al., (2024)](https://openreview.net/forum?id=upAWnMgpnH), [Longpre et al. (2024)](https://openreview.net/forum?id=tH1dQH20eZ), and [Bommasani et al., (2025)](https://openreview.net/forum?id=x6fXnsM9Ez).
> - Finally, many ML researchers contribute to work that is relevant to governance and policy—such as interpretability, robustness, and auditing. These are core topics in our taxonomy, and TMLR’s readership includes precisely this community. We therefore take the view that this work will be relevant to much of TMLR’s audience.
>
>
> > Include reference to https://arxiv.org/abs/2401.15897 in assessment paragraph 2 - it's really just an excellent paper and belongs amid the references
>
> We thank the reviewer for this suggestion. The authors are aware of this paper and agree it is relevant to this section. We will add it as a citation in Section 3 as suggested.
>
> We would like to thank the reviewer again for their time and expert input, and hope that the above has allayed their concerns.

---

### Author Response · Authors · 2025-03-05
**Revision uploaded incorporating reviewer feedback**

Dear reviewers,

Thanks for your feedback on the paper. Following reviews, we have now uploaded a revision of the paper incorporating suggestions and feedback. Changes to the previous version are made in red text for visibility.

Thank you once again.

---

### Decision · Action_Editor_Pd9u · 2025-03-26

**Recommendation:** Accept as is

**Comment:**

The manuscript provides a broad survey of TAIG issues and would be an excellent on-ramp for researchers new to the field. Reviewer are positive about the work after authors made revisions to add procedural detail and better convey some points.

**Audience:**

All reviewers and the AE agree there is certainly an audience for this work.

**Claims And Evidence:**

After making appropriate revisions to the manuscript, all reviewers agree that the claims are well-backed by appropriate literature and a description of the methodology for creating the survey.